

# An efficient and straightforward online vector quantization method for a data stream through remove-birth updating

Kazuhisa Fujita

Komatsu University, Komatsu, Ishikawa, Japan

## ABSTRACT

The growth of network-connected devices has led to an exponential increase in data generation, creating significant challenges for efficient data analysis. This data is generated continuously, creating a dynamic flow known as a data stream. The characteristics of a data stream may change dynamically, and this change is known as concept drift. Consequently, a method for handling data streams must efficiently reduce their volume while dynamically adapting to these changing characteristics. This article proposes a simple online vector quantization method for concept drift. The proposed method identifies and replaces units with low win probability through remove-birth updating, thus achieving a rapid adaptation to concept drift. Furthermore, the results of this study show that the proposed method can generate minimal dead units even in the presence of concept drift. This study also suggests that some metrics calculated from the proposed method will be helpful for drift detection.

## INTRODUCTION

In today's world, an enormous number of devices, from computers to Internet of Things (IoT) gadgets, are constantly connected to the Internet, sending a continuous stream of data to server computers. A continuous generation of data that flows like a river into a server is called a data stream (*Ding et al., 2015*). Since the volume of an accumulated data stream is too large to store (*Gama et al., 2014*), we need preprocessing to extract its synopsis, such as vector quantization, to handle it effectively (*Alothali, Alashwal & Harous, 2019*). This article proposes a new approach of an online vector quantization method for a data stream to reduce the data size and store the features of a data stream.

In many real-world scenarios, we cannot assume that a dataset is static (*Ramírez-Gallego et al., 2017*). Instances of a data stream will evolve (*Zubaroğlu & Atalay, 2021*), meaning that not only is the data continuously generated, but its properties can also change. This phenomenon is known as concept drift. A quantization method for concept drift must be able to extract new features as the characteristics of the data stream change. Therefore, we need a method that can resize the data and automatically adapt to changes in the characteristics of the data stream.

Corresponding author
Kazuhisa Fujita,
kazu@spikingneuron.net

Developing a machine learning method for data streams with concept drift is a new computational challenge (*Sultan, 2022*) because the method needs to fulfill four capabilities: avoidance of storing data points, feature extraction, flexibility, and single pass. The method should avoid storing whole data points (*Beyer & Cimiano, 2013*) because the accumulated data is enumerated, so the method must extract the features of the data and store them instead. The distribution of a data stream will evolve, which can degrade the model's performance if it does not adapt to these changes. Therefore, the method should be able to flexibly extract new features on the fly. In addition, since data streams are potentially unbounded and ordered sequences of instances (*Zubaroğlu & Atalay, 2021*), the method should continuously update the extracted features or the model with new data using the latest information. Batch algorithms are not ideal for dynamically changing data, so the algorithm must be capable of continuous online (single-pass) learning (*Smith & Alahakoon, 2009*).

This article proposes improved online vector quantization methods for a data stream. Our methods are designed to quickly adapt to the evolution of the data stream with concept drift. Specifically, this study focus on remove-birth (RB) updating to improve online quantization methods for concept drift.

The inspiration for the RB updating mechanism is derived from the death-birth updating concept originally proposed by *Ohtsuki et al. (2006)* for an evolutionary game on a graph. This concept revolves around the random selection and elimination of an individual from a node, creating an empty node. A new individual is then introduced to fill this node, inheriting the characteristics or strategies of a neighboring node.

RB updating consists of removing units (reference vectors) far from the current data distribution and creating new units around the units on the distribution. This procedure results in a simple and efficient online vector quantization method for data streams. Using a reliable metric to determine whether a unit should be removed is critical for RB updating. In this study, the win probability of a unit is used as the metric. Notably, the win probability is not affected by the value range of data. As a result, using the win probability as the metric allows us to efficiently decide whether to remove a unit even when the value range of the data changes due to concept drift.

This study proposes three simple online vector quantization methods for data streams, which are online k-means (*MacQueen, 1967*), Kohonen's self-organizing maps (SOM) (*Kohonen, 1982*), and neural gases (NG) (*Martinetz & Schulten, 1991*) applied RB updating, named online k-means with RB updating (OKRB), SOM with RB updating (SOMRB), and NG with RB updating (NGRB), respectively. Online k-means is an online version of k-means. SOM is a competitive learning method used for cluster analysis, data visualization, and dimensionality reduction. NG, an alternative to SOM, is used for vector quantization and data representation as a graph. All three methods are online (incremental) learning techniques suitable for data streams. However, online learning alone is not sufficient to handle data streams, which need mechanisms to forget outdated data and adapt to the latest state of nature (*Gama, 2010*). Online k-means, SOM, and NG need to be improved because their parameters decay with iterations and they cannot adapt to the latest state of the data. This problem is resolve by RB updating. This study shows that

OKRB, SOMRB, and NGRB exhibit satisfactory performance in vector quantization and can quickly adapt to concept drift. Thus, the proposed methods can adapt to concept drift by using RB updating. Portions of this text were previously published as part of a preprint (https://export.arxiv.org/abs/2306.12574).

## CONCEPT DRIFT

In most real-world applications, a data stream is not strictly stationary, which means that its concept could change over time (*Gama, 2010*). This unanticipated evolution of the statistical properties of a data stream is known as concept drift. Concept drift refers to situations where the underlying patterns or distributions of the data stream evolve, resulting in unexpected changes in the statistical properties (*Zubaroğlu & Atalay, 2021*).

*Ramírez-Gallego et al. (2017)* have classified concept drift into four types: sudden, gradual, incremental, and recurring. Sudden concept drift refers to sudden changes in the statistical properties of a data stream. Gradual concept drift is characterized by an evolution in which the number of data points generated by a data stream with previous properties gradually decreases, while those with new properties gradually increase over time. Incremental concept drift involves the step-by-step transformation of the statistical properties of a data stream. Recurring concept drift involves the cyclical change of a data stream between two or more characteristics. More details about these types of concept drift can be found in *Zubaroğlu & Atalay (2021)*.

## RELATED WORK

K-means (*MacQueen, 1967*; *Lloyd, 1982*) is the simplest and most well-known clustering method. K-means has gained widespread recognition and is considered one of the top ten algorithms used in data mining (*Wu et al., 2007*). Its popularity is due to its ease of implementation and efficient performance (*Haykin, 2009*). K-means is also a useful vector quantization method because it transforms a dataset into a set of centroids. The typical k-means algorithm is the Lloyd algorithm (*Lloyd, 1982*), a well-known instance of k-means that uses batch learning. Online k-means, also known as sequential k-means or MacQueen's k-means (*MacQueen, 1967*), uses online learning. In particular, online k-means can be applied to quantization for data streams because it is not limited by the size of the data using online learning.

The most famous and widely used self-organizing map algorithm is Kohonen's self-organizing map (SOM) (*Kohonen, 1982*). This method represents an input dataset as units with a weight vector called the reference vector. The SOM can project multidimensional data onto a low-dimensional map (*Vesanto & Alhoniemi, 2000*). More specifically, the low-dimensional map is typically a two-dimensional grid because it is easy to visualize the data (*Smith & Alahakoon, 2009*). SOM is essentially to perform topological vector quantization of the input data to reduce the dimensionality of the input data while preserving as much of the spatial relationships within the data as possible (*Smith & Alahakoon, 2009*). Because of these capabilities, SOM is widely used in data mining, especially in unsupervised clustering (*Smith & Alahakoon, 2009*). The versatility of SOM

is demonstrated by its use in a wide range of applications, such as skeletonization (*Singh, Cherkassky & Papanikolopoulos, 2000*), data visualization (*Heskes, 2001*), and color quantization (*Chang et al., 2005*; *Rasti, Monadjemi & Vafaei, 2011*).

Neural gas (NG), proposed by *Martinetz & Schulten (1991)*, is one of the SOM alternatives. With the ability to quantize input data and generate reference vectors, NG constructs a network that reflects the manifold of the input data. One of its good features is its independence from the range of values of data. However, NG needs to improve when dealing with data streams. The root of NG's problems with data streamslies in its time-decaying parameters. While this decay helps NG's network fit the input data more accurately, it reduces its flexibility over time. Thus, in scenarios where the characteristics of the data suddenly change during training, NG cannot adjust its network accordingly. This inherent inflexibility makes NG unsuitable for data stream applications. In addition, maintaining static parameters in NG will result in the creation of dead units that have no nearest data point. In particular, NG with non-decaying parameters tends to create dead units when reference vectors are distant from the data. These characteristics pose significant barriers to the application of NG to data streams.

Growing neural gas (GNG) (*Fritzke, 1994*) is also a kind of SOM alternative (*Fisšer, Faigl & Kulich, 2013*) and can find the topology of an input distribution (*García-RodríGuez et al., 2012*). GNG can also quantize input data and create reference vectors from data. GNG changes the network structure by increasing the number of neurons during training. This makes the network remarkably flexible and reflective of the data structure. GNG has a wide range of applications, including topology learning, such as landmark extraction (*Fatemizadeh, Lucas & Soltanian-Zadeh, 2003*), cluster analysis (*Canales & Chacón, 2007*; *Costa & Oliveira, 2007*; *Fujita, 2021*), reconstruction of 3D models (*Holdstein & Fischer, 2008*), object tracking (*Frezza-Buet, 2008*), extraction of the two-dimensional outline of an image (*Angelopoulou, Psarrou & García-Rodríguez, 2011*; *Angelopoulou et al., 2018*), and anomaly detection (*Sun, Liu & Harada, 2017*).

The vector quantization methods discussed previously, including k-means, SOM, NG, and GNG, are effective for static data processing. However, they face challenges when applied to non-stationary data, such as data streams. To overcome this limitation, much research has been devoted to developing both clustering and vector quantization methods specifically designed for data streams.

Several stream k-means methods have been proposed. Incremental k-means (*Ordonez, 2003*) is a binary data clustering method based on k-means. StreamKM++ (*Ackermann et al., 2012*) is based on k-means++ (*Arthur & Vassilvitskii, 2007*), which improves the initial centroid problem of k-means. These methods can store sufficient statistics and extract features from the entire dataset, even though the methods use a portion of the data at each step. However, these methods are designed to cluster large datasets on a computer with small memory, and they do not assume that the characteristics of the data streams change during training. Thus, they may not be able to capture the feature of a data stream with a new property because they retain the statistical properties of the old data.

Several researchers have improved self-organizing maps (SOM) and growing neural gas (GNG) for data stream analysis. Smith's GCPSOM (*Smith & Alahakoon, 2009*) is an

online SOM algorithm suitable for large datasets. It incorporates the forgetting factor and the growing network that allows the network to let go of old patterns and adapt to new ones as they emerge. However, the increased flexibility and adaptability come at the cost of increased complexity and a larger parameter set than traditional Kohonen's SOM. *Ghesmoune, Azzag & Lebbah (2014)* and *Ghesmoune, Lebbah & Azzag (2015)* have proposed G-stream, an adaptation of GNG designed for data stream clustering. G-stream incorporates a reservoir to accumulate outliers and stores the timestamps of data points that belong to a time window. Although these features enhance the precision of the model, they introduce more hyperparameters. *Silva & Marques (2015)* have introduced a variant of SOM for data streams, called ubiquitous SOM, which uses the average quantization error over time to estimate learning parameters. This approach allows the model to maintain continuous adaptability and handle concept drift in multidimensional streams. The output is a meticulously distributed two-dimensional grid, indicative of the algorithm's sophisticated data mapping capabilities. However, ubiquitous SOM assumes that all attributes are normalized. This may undermine its effectiveness in scenarios with significant fluctuations in data value ranges.

Data stream analysis often uses a two-step approach consisting of an online phase followed by an offline phase. In the online phase, data is assigned to microclusters, which are a summary of the data. This is the data abstraction step. The offline phase produces the clustering result. An example of this method is CVD-stream (*Mousavi et al., 2020*), an online-offline density-based clustering method for data streams.

Many researchers have attempted to improve vector quantization methods for data streams. However, these improvements often introduce additional hyperparameters, increasing the complexity of the methods. Methods using a growing network can efficiently adapt to a data stream but not maintain a static number of units unless network pruning is performed. Their applications are limited because a postprocessing method such as spectral clustering often requires a fixed number of units (reference vectors). In contrast, our proposed method blends simplicity with adaptability to a data stream using RB updating. RB updating focuses on the win probabilities of units to decide which unit to remove or add. We improve the original vector quantization methods with only two additional hyperparameters using RB updating to make them adaptable to a data stream. RB updating allows for easy implementation and tuning of the proposed method.

## ONLINE QUANTIZATION METHODS WITH REMOVE-BIRTH UPDATING

In this study, remove-birth updating (RB updating) is applied to three different online quantization methods: online k-means, SOM, and NG. This approach allows us to quickly adapt to changes in the distribution of the data. Online k-means is a type of k-means that uses online learning, while SOM can project a low-dimensional space. NG can extract the topology of the input as a graph. By integrating RB updating with these methods, we will achieve data quantization, dimension reduction, and topology extraction for a data stream.

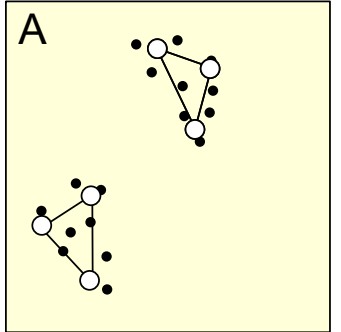 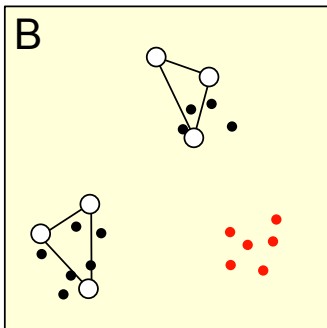 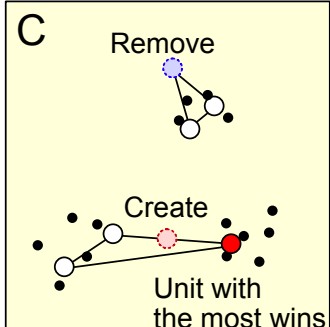

**Figure 1** **This figure provides a schematic representation of the remove birth (RB) updating process.** Each dot represents a data point, while the open circles represent units. (A) The initial distribution of data points and their corresponding units. (B) The change in data distribution and the introduction of new data points, marked in red, resulting from concept drift. At the same time, the units retain their positions from the initial data distribution. (C) The removal of the least frequent winning unit, marked by the blue dotted circle. A new unit, marked by the red dotted circle, is then introduced around the most frequent winning unit, marked by the red-filled circle.

## Metric for RB updating

Concept drift results in a change in the characteristics of a data stream. As a result of this change, some units representing the data with old characteristics may be outside the data distribution with new characteristics, as shown in Figs. 1A and 1B. RB updating addresses this problem by removing units that are far from the current data distribution and creating new units around the units on the distribution, as shown in Fig. 1C. Developing an effective metric for determining when and which unit to remove and where to create a unit is critical to RB updating.

*Fritzke (1997)* have proposed a metric based on error, which is the difference (distance) between a data point and its corresponding reference vector. However, the effectiveness of this error-based metric is affected by the value range of the dataset. Thus, this metric will face problems for data streams undergoing concept drift because there is no guarantee that the value range is static for data streams.

This study introduces win probability as an alternative metric. A win of a unit means that the unit is closest to an input data point. Unlike error-based metrics, win probability retains independence from the value range of the dataset, thereby enhancing its effectiveness in RB updating even as the value range of the dataset changes due to concept drift. This metric $M_n(t)$ of unit $n$ is expressed as

$$M_n(t) = \frac{P_{\text{win},n}(t)}{\max_n P_{\text{win},n}(t)} = \frac{c_n(t)}{\max_n c_n(t)}. \tag{1}$$

Here, $P_{\text{win},n}(t) = c_n(t)/(\sum_m c_m(t))$ refers to the win probability of unit $n$, and $c_n(t)$ refers to the number of wins of unit $n$. In this study, $c_n(t)$ shows the decay over iteration as indicated:

$$c_n(t+1) = c_n(t) - \beta c_n(t), \tag{2}$$

where $\beta$ is a decay constant. This exponential decay mechanism limits the influence of old data characteristics, thus improving the overall effectiveness of the RB updating process.

## Online k-means with RB updating

Online k-means with RB updating (OKRB) is a modification of the standard online k-means algorithm designed to quickly adapt to variations within a data stream. However, online k-means suffers from the problem of dead units. Dead units are units (centroids) to which no data points are assigned, often due to their initial placement far from the input dataset. When concept drift occurs, and results in units far from the data distribution, units close to the dataset may gradually move toward it, while those farther away remain static. When concept drift occurs and the data distribution changes, online k-means will generate dead units. OKRB mitigates this shortcoming through RB updating. The detailed procedure for the OKRB algorithm is described in Algorithm 1.

Consider an input data sequence denoted by $X = \{x_1, x_2, \ldots, x_t, \ldots\}$, where each $x_t$ belongs to a $D$-dimensional real space $\mathbb{R}^D$. The probability distribution generating $X$ may change during data point generation due to concept drift.

OKRB contains $N$ units, where each unit represents a centroid. Each unit $i$ is associated with a reference vector $w_n \in \mathbb{R}^D$. To accommodate evolving data streams, OKRB iteratively adapts its reference vectors to the incoming data point $x_t$, eliminating less useful units through RB updating.

In the initialization phase described in Algorithm 1 (steps 2 and 3), we initialize the reference vector $w_n$ and the win count $c_n$. Specifically, each reference vector $w_n$ is constructed according to the following equation:

$$w_n = (\xi_1, \ldots, \xi_d, \ldots, \xi_D), \tag{3}$$

where each component $\xi_d$ is sampled uniformly at random from the interval $[0, 1)$. The win count $c_n$ is initialized to zero.

In each iteration, OKRB processes a single data point $x_t$. Initially, the algorithm receives $x_t$ (Algorithm 1, step 5) and identifies the winning unit $n_1$, which is the unit whose reference vector is closest to $x_t$ (Algorithm 1, step 7). The reference vector of $n_1$ is updated by

$$w_{n_1} \leftarrow w_{n_1} + \varepsilon(x_t - w_{n_1}), \tag{4}$$

where $\varepsilon$ is the learning rate (Algorithm 1, step 8). Unlike traditional online k-means, where the learning rate decays over time, in OKRB the learning rate remains static. This is because the data streams are unbounded, making the end of the iteration indeterminable.

RB updating in OKRB serves to prune less winning units while introducing new units near frequently winning units. Specifically, each iteration $n_1$'s win count $c_{n_1}$ is incremented by one (Algorithm 1, step 10). The algorithm then identifies the unit with the maximum number of wins $n_{\max}$ (Algorithm 1, step 11) and the unit with the minimum number of wins $n_{\min}$ (Algorithm 1, step 12). If $c_{n_{\min}}/c_{n_{\max}}$ exceeds a certain threshold $\text{TH}_{\text{RB}}$, RB updating is triggered (Algorithm 1, step 13): the unit $n_{\min}$ is discarded and a new unit $n_{\text{new}}$ is added near $n_{\max}$. In Algorithm 1, $n_{\text{new}} = n_{\min}$. The reference vector and the win count of $n_{\text{new}}$ are determined by $w_{n_{\text{new}}} = (w_{n_{\max}} + w_f)/2$ and $c_{n_{\text{new}}} = (c_{n_{\max}} + c_f)/2$, respectively, where $f$ is the unit closest to $n_{\max}$ (Algorithm 1, steps 14–16).

Finally, the winning counts of all units are exponentially decaying (Algorithm 1, step 18), according to

$$c_n \leftarrow c_n - \beta c_n, \tag{5}$$

with $\beta$ as the decay rate.

---

**Algorithm 1** Online k-means with RBirth updating (OKRB)

---

**Require:** $X = \{\boldsymbol{x}_1, \ldots, \boldsymbol{x}_t, \ldots\}, N$

1: **Initialize:**
2:     $\boldsymbol{w}_n = (\xi_1, \ldots, \xi_d, \ldots, \xi_D), n=1, \ldots, N$, where $\xi_d = [0,1)$ is uniformed random
3:     $c_n = 0, n=1, \ldots, N$
4: **loop**
5:     Receive input $\boldsymbol{x}_t$ at iteration $t$
6:     {Update reference vectors:}
7:     $n_1 = \arg \min_n \|\boldsymbol{x}_t - \boldsymbol{w}_n\|$
8:     $\boldsymbol{w}_{n_1} \leftarrow \boldsymbol{w}_{n_1} + \varepsilon(\boldsymbol{x}_t - \boldsymbol{w}_{n_1})$
9:     {RB updating:}
10:     $c_{n_1} \leftarrow c_{n_1} + 1$
11:     $n_{\max} = \arg \max_n c_n$
12:     $n_{\min} = \arg \min_n c_n$
13:     **if** $c_{n_{\min}}/c_{n_{\max}} < \mathrm{TH_{RB}}$ **then**
14:         $f = \arg \min_n \|\boldsymbol{w}_n - \boldsymbol{w}_{n_{\max}}\|$
15:         $\boldsymbol{w}_{n_{\min}} = (\boldsymbol{w}_{n_{\max}} + \boldsymbol{w}_f)/2.$
16:         $c_{n_{\min}} = (c_{n_{\max}} + c_f)/2.$
17:     **end if**
18:     $c_n \leftarrow c_n - \beta c_n, n=1, \ldots, N$
19: **end loop**

---

## SOM with RB updating

Self-Organizing Map with RB updating (SOMRB) is based on standard SOM, but with improved adaptation to changes in the data stream through RB updating. SOMRB projects high-dimensional input data onto a low-dimensional map that can be used for clustering, visualization, and dimension reduction. The detailed SOMRB algorithm is given in Algorithm 2.

SOMRB contains $N$ units, each with a reference vector $\boldsymbol{w}_n$ in $\mathbb{R}^d$. These units, positioned at $\boldsymbol{p}_n = (p_{n1}, p_{n2}) \in \mathbb{R}^2$ on a two-dimensional map, form a grid structure with $p_{n1}$ and $p_{n2}$ as integers. First, unit $n$ is positioned at $\boldsymbol{p}_n = (\lfloor n/L \rfloor, (\mathrm{mod}L))$, where $L = \lfloor \sqrt{N} \rfloor$ (Algorithm 2, steps 2 and 3). Units are connected to their nearest neighbors, forming a 2-dimensional grid (Algorithm 2, step 4). The reference vector is initialized as $\boldsymbol{w}_n = (n/L, (\mathrm{mod}L)/L, 0, \ldots, 0)$ (Algorithm 2, step 5). Such an initialization strategy mitigates the distortion of SOMRB's mesh topology. The win count $c_n$ are set to 0 (Algorithm 2, step 6).

In each iteration, SOMRB receives a single data point $x_t$ (Algorithm 2, step 8) and adjusts its reference vectors. The winning unit, $n_1$, with the reference vector closest to $x_t$, is identified (Algorithm 2, step 10). Then, all reference vectors are updated according to

$$w_n \leftarrow w_n + \varepsilon h(\|p_n - p_{n_1}\|)(x_t - w_n), \tag{6}$$

where $\varepsilon$ is the learning rate and $h(\cdot)$ is the neighborhood function defined as $h(d) = \exp(-\frac{d^2}{2\sigma^2})$ (Algorithm 2, step 11).

RB updating is used to remove units that rarely win (dead units) and to add new units around units that frequently win. The win count of unit $n_1$, denoted by $c_{n_1}$, is incremented at each iteration (Algorithm 2, step 13). The unit with the maximum count $c_{n_{max}}$ is identified from the units with neighboring empty vertices (Algorithm 2, steps 14 and 15). Note that $n_{max}$ must have neighboring empty vertices on the grid. The number of edges of the unit with neighboring empty vertices $e_n$ is less than four, because a unit on the grid can have a maximum of four edges (Algorithm 2, step 14). The unit with the minimum count $c_{n_{min}}$ is also identified (Algorithm 2, step 16). When $c_{n_{min}}/c_{n_{max}}$ exceeds the threshold $\text{TH}_{RB}$, an RB update is performed (Algorithm 2, step 17). This involves removing the minimum winning unit $n_{min}$ (Algorithm 2, step 18) and adding a new unit $n_{new}$ near the maximum winning unit $n_{max}$ (Algorithm 2, step 19). In Algorithm 2, $n_{new} = n_{min}$. $n_{new}$ is placed on an empty vertex neighboring $n_{max}$, which is chosen randomly if there is more than one empty vertex. $n_{new}$ is connected to the neighboring units on the grid (Algorithm 2, step 20). The reference vector $w_{n_{new}}$ of $n_{new}$ is computed based on the average of the reference vectors of the neighboring units if it has more than one neighbor (Algorithm 2, steps 21–23). $c_{n_{new}}$ is the average win count of the neighboring units. Conversely, if $n_{new}$ has only one neighbor (*i.e.*, $n_{new}$ connects only to $n_{max}$), the reference vector of $n_{new}$ is the average of the reference vectors of $n_{max}$ and its neighbors (Algorithm 2, steps 26–29). In scenarios in which $n_{max}$ connects only to $n_{new}$, the reference vector $w_{n_{new}}$ is the average of the reference vectors of $n_{max}$ and its nearest unit $f$ (Algorithm 2, steps 30–33).

All winning counts are exponentially decaying according to the following formula (Algorithm 2, step 37):

$$c_n \leftarrow c_n - \beta c_n, \tag{7}$$

where $\beta$ is the decay rate.

## Neural gas with RB updating

Neural gas with remove-birth updating (NGRB) is an alternative to Self-Organizing Map (SOM) for data streams based on neural gas (NG). NGRB quantizes the input data and generates a network like NG. In addition, NGRB can reconfigure its network structure more quickly through RB updating. The complete NGRB algorithm is given in Algorithm 3.

The network generated by NGRB consists of $N$ units, with edges connecting pairs of units. Each unit $i$ has a reference vector $w_i$ and a winning counter $c_n$. Edges between units are neither weighted nor directed. The edge is represented by $C_{nm}$, which is 0 or 1. If $C_{nm} = 1$, unit $n$ is connected to unit $m$, and vice versa. In Algorithm 3, $C_{nm} = C_{mn}$. Each edge has an age variable, denoted by $a_{nm}$, which informs the decision to keep or discard

---

**Algorithm 2** Self-Organizing Map with RB updating (SOMRB)

---

**Require:** $X = \{\boldsymbol{x}_1, \ldots, \boldsymbol{x}_t, \ldots\}, N$

1: **Initialize:**

2:     $L = \lfloor \sqrt{N} \rfloor$

3:     $\boldsymbol{p}_n = (\lfloor n/L \rfloor, (n \bmod L)), \text{n=0}, \ldots, N-1$

4:     Connect each pair of neighboring units

5:     $\boldsymbol{w}_n = (\lfloor n/L \rfloor/L, (n \bmod L)/L, 0, \ldots, 0), \text{n=0}, \ldots, N-1$

6:     $c_n = 0, \text{n=0}, \ldots, N-1$

7: **loop**

8:     Receive input $\boldsymbol{x}_t$ at iteration $t$

9:     {Update reference vectors:}

10:     $n_1 = \arg\min_n \|\boldsymbol{x}_t - \boldsymbol{w}_n\|$

11:     $\boldsymbol{w}_n \leftarrow \boldsymbol{w}_n + \varepsilon h(\|\boldsymbol{p}_n - \boldsymbol{p}_{n_1}\|)(\boldsymbol{x} - \boldsymbol{w}_n), \text{n=0}, \ldots, N-1$

12:     {RB updating:}

13:     $c_{n_1} \leftarrow c_{n_1} + 1$

14:     $M = \{n | e_n < 4\}$, where $e_n$ is the number of edges connected to unit $n$

15:     $n_{\max} = \arg\max_{n \in M} c_n$

16:     $n_{\min} = \arg\min_n c_n$

17:     **if** $c_{n_{\min}}/c_{n_{\max}} < \text{TH}_{\text{RB}}$ **then**

18:         Remove unit $n_{\min}$

19:         Add new unit $n_{\min}$ on the empty vertex neighboring unit $n_{\max}$

20:         Establish edges between unit $n_{\min}$ and its neighboring units

21:         $M_{\min} = \{n | n \text{ is a neighbor of } n_{\min}\}$

22:         **if** $|M_{\min}| > 1$ **then**

23:             $\boldsymbol{w}_{n_{\min}} = \frac{1}{|M_{\min}|} \sum_{n \in M_{\min}} \boldsymbol{w}_n.$

24:             $c_{n_{\min}} = \frac{1}{|M_{\min}|} \sum_{n \in M_{\min}} c_n.$

25:         **else**                     ▷ $|M_{\min}| = 1$ and unit $n_{\min}$ connects with only unit $n_{\max}$

26:             $M_{\max} = \{n | n \text{ is a neighbor of } n_{\max} \text{ excluding } n_{\min}\} \cup \{n_{\max}\}$

27:             **if** $|M_{\max}| > 1$ **then**

28:                 $\boldsymbol{w}_{n_{\min}} = \frac{1}{|M_{\max}|} \sum_{n \in M_{\max}} \boldsymbol{w}_n.$

29:                 $c_{n_{\min}} = \frac{1}{|M_{\max}|} \sum_{n \in M_{\max}} c_n.$

30:             **else**       ▷ $|M_{\max}| = 1$ and unit $n_{\max}$ connects with only unit $n_{\min}$

31:                 $f = \arg\min_n \|\boldsymbol{w}_n - \boldsymbol{w}_{n_{\max}}\|$

32:                 $\boldsymbol{w}_{n_{\min}} = (\boldsymbol{w}_{n_{\max}} + \boldsymbol{w}_f)/2.$

33:                 $c_{n_{\min}} = (c_{n_{\max}} + c_f)/2.$

34:             **end if**

35:         **end if**

36:     **end if**

37:     $c_n \leftarrow c_n - \beta c_n, \text{n=0}, \ldots, N-1$

38: **end loop**

---

the edge. In NGRB, data points are represented iteratively, and the reference vectors are refined at each iteration.

In the initialization phase described in Algorithm 3 (steps 2-4), we initialize the reference vector $w_n$, the win count $c_n$, and $C_{nm}$. Specifically, each reference vector $w_n$ is constructed according to the following equation:

$$w_n = (\xi_1, \ldots, \xi_d, \ldots, \xi_D), \tag{8}$$

where each component $\xi_d$ is sampled uniformly at random from the interval $[0, 1)$. $c_n$ and $C_{nm}$ is initialized to zero.

In each iteration, NGRB processes a single data point. After receiving $x_t$ (Algorithm 3, step 6), NGRB refines its reference vectors and changes the network topology. The reference vector update procedure is identical to that of NG. We determine the neighborhood ranking $k_n$ of unit $n$ based on the distance between $w_n$ and $x_t$ (Algorithm 3, step 8). This results in a sequence of unit rankings $(n_0, n_1, \ldots, n_k, \ldots, n_{M-1})$ determined by

$$\|x - w_{n_0}\| < \|x - w_{n_1}\| < \ldots < \|x - w_{n_k}\| < \ldots < \|x - w_{n_{N-1}}\|. \tag{9}$$

Then, the reference vectors of all units are updated according to

$$w_n \leftarrow w_n + \varepsilon e^{-k_n/\lambda}(x_t - w_n), n = 1, \ldots, N, \tag{10}$$

where $e^{-k_n/\lambda}$ is a neighborhood function (Algorithm 3, step 9). $\lambda$ determines the number of units that significantly change their reference vectors at each iteration.

At the same time, the network topology evolves in response to the input data through iterative adaptation. The adaptation procedure of the network topology is also identical to that of NG. The winning unit $n_0$ and the second nearest unit $n_1$ are identified. If $C_{n_0 n_1} = 0$, we set $C_{n_0 n_1} = 1$ and $a_{n_0 n_1} = 0$ (Algorithm 3, steps 11–13). If $C_{n_0 n_1} = 1$, we set $a_{n_0 n_1} = 0$ (Algorithm 3, steps 14 and 15). The age of all edges connected to $n_0$ is incremented by 1 (Algorithm 3, step 17), and links exceeding the prescribed lifetime are removed (Algorithm 3, step 18).

In addition, NGRB dynamically restructures its network using RB updating, which involves removing infrequently winning units and introducing new units around those that win frequently. The win count of the winning unit, denoted by $c_{n_0}$, is incremented at each iteration (Algorithm 3, step 20). The algorithm then identifies the unit with the maximum wins $n_{\max}$, and the unit with the minimum wins $n_{\min}$ (Algorithm 3, steps 21 and 22). When $c_{n_{\min}}/c_{n_{\max}}$ exceeds the threshold $TH_{RB}$, $n_{\min}$ is eliminated and a new unit $n_{new}$ is added around $n_{\max}$ (Algorithm 3, steps 23–25). In Algorithm 3, $n_{new} = n_{\min}$. The reference vector and the win count of $n_{new}$ are respectively determined by $w_{n_{new}} = (w_{n_{\max}} + w_f)/2$ and $c_{n_{new}} = (c_{n_{\max}} + c_f)/2$, where $f$ is the neighboring unit of $n_{\max}$ that has the maximum wins (Algorithm 3, steps 26-33). If $n_{\max}$ has no neighbors, the closest unit to $n_{\max}$ is taken as unit $f$ (Algorithm 3, step 30). Consequently, $n_{new}$ is connected to $n_{\max}$ and $f$, setting $C_{n_{new} n_{\max}} = 1$ and $C_{n_{new} n_f} = 1$ (Algorithm 3, steps 34). The ages of these new edges, $a_{n_{\min} n_{\max}}$ and $a_{n_{\min} f}$, are set to 0 (Algorithm 3, step 35).

All winning counters are subject to exponential decay, calculated as follows

$$c_n \leftarrow c_n - \beta c_n, \tag{11}$$

where $\beta$ is the decay rate (Algorithm 3, step 37).

---

**Algorithm 3** Neural Gas with RB updating (NGRB)

---

**Require:** $X = \{x_1, \ldots, x_t, \ldots\}, N$

1: **Initialize:**
2:     $w_n \leftarrow (\xi_1, \ldots, \xi_d, \ldots, \xi_D), \text{n}=1, \ldots, N$, where $\xi_d = [0, 1)$ is uniformed random value
3:     $c_n \leftarrow 0, \text{n}=1, \ldots, N$
4:     $C_{nm} \leftarrow 0$ for all $n = 1$ to $N$ and $m = 1$ to $N$
5: **loop**
6:       Receive input $x_t$ at iteration $t$
7:       {Adaptation of reference vectors:}
8:       Determine the neighborhood-ranking $k_n$
9:       $w_n \leftarrow w_n + \varepsilon e^{-k_n/\lambda}(x_n - w_n), \text{n}=1, \ldots, N$
10:      {Training the network topology:}
11:      **if** $C_{n_0 n_1} = 0$ **then**
12:         $C_{n_0 n_1} = 1$
13:         $a_{n_0 n_1} = 0$
14:      **else**
15:         $a_{n_0 n_1} = 0$
16:      **end if**
17:      Increment all the ages of the edge emerging from the unit $n_1$
18:      Remove the edge with $a_{nm} > a_{\max}$
19:      {RB updating:}
20:      $c_{n_0} \leftarrow c_{n_0} + 1$
21:      $n_{\max} = arg \; \max_n c_n$
22:      $n_{\min} = arg \; \min_n c_n$
23:      **if** $c_{n_{\min}}/c_{n_{\max}} < \text{TH}_{\text{RB}}$ **then**
24:         Remove unit $n_{\min}$
25:         Create a new unit at $n_{\min}$
26:         $M = \{n | n \text{ is a neighbor of } n_{\max}\}$
27:         **if** $|M| > 0$ **then**
28:            $f = arg \; \max_{n \in M} c_n$
29:         **else**
30:            $f = arg \; \min_{n \neq n_{\max}} \|w_n - w_{n_{\max}}\|$
31:         **end if**
32:         $w_{n_{\min}} = (w_{n_{\max}} + w_f)/2.$
33:         $c_{n_{\min}} = (c_{n_{\max}} + c_f)/2.$
34:         Connect $n_{\min}$ with $n_{\max}$ and $f$
35:         $a_{n_{\min} n_{\max}} = a_{n_{\min} f} = 0$
36:      **end if**
37:      $c_n \leftarrow c_n - \beta c_n, \text{n}=1, \ldots, N$
38: **end loop**

---

**Table 1  Parameters for OKRB, OKRB using the error-based metric (OKRB EB-based), and online k-means.**

|  | OKRB | OKRB EB-based | Online k-means |
|---|---|---|---|
| $\varepsilon$ | 0.1 | 0.3 | 0.5 |
| $TH_{RB}$ | 0.01 | 0.01 | – |
| $\beta$ | 0.005 | 0.005 | – |

**Table 2  Parameters for SOMRB, SOMRB using the error-based metric (SOMRB EB-based), and SOM.**

|  | SOMRB | SOMRB EB-based | SOM |
|---|---|---|---|
| $\varepsilon$ | 0.2 | 0.3 | 0.4 |
| $\sigma$ | 0.5 | 0.5 | 0.5 |
| $TH_{RB}$ | 0.1 | 0.5 | – |
| $\beta$ | 0.0001 | 0.0005 | – |

**Table 3  Parameters for NGRB, NGRB using the error-based metric (NGRB EB-based), and NG.**

|  | NGRB | NGRB EB-based | NG |
|---|---|---|---|
| $\varepsilon$ | 0.3 | 0.3 | 0.3 |
| $\lambda$ | 0.5 | 1 | 2 |
| $a_{max}$ | 75 | 100 | 75 |
| $TH_{RB}$ | 0.01 | 0.01 | – |
| $\beta$ | 0.005 | 0.005 | – |

## Parameters

In this study, online k-means, SOM, NG, and GNG are used as comparison methods. To investigate the efficiency of the win probability based metric on RB updating, OKRB, SOMRB, and NGRB using the error-based metric were evaluated. For details on RB updating using the error-based metric, see 'RB updating using error-based metric' in the Appendix. The parameters for all methods except $N = 100$ are determined using a grid search technique, details of which can be found in 'Parameter optimization' in the Appendix. The parameters for OKRB, OKRB using the error-based metric, and online k-means are shown in Table 1. The parameters for SOMRB, SOMRB using the error-based metric, and SOM are shown in Table 2. The parameters for NGRB, NGRB using the error-based metric, and NG are shown in Table 3. For GNG, the following parameters were used: $\varepsilon_w = 0.1$, $\varepsilon_n = 0.0005$, $\lambda = 50$, $\alpha = 0.25$, $\beta = 0.999$, and $a_{max} = 25$.

## EXPERIMENTAL SETUP

### Comparison methods

The proposed methods are compared with four other methods: online k-means, SOM, NG, and GNG. The algorithm of online k-means is referred to in the Appendix section labeled 'Online k-means'. The detailed descriptions of SOM, NG, and GNG are covered in *Fujita (2021)*. Notably, online k-means, SOM, and NG have parameters that decay over iterations, but in this study they were kept static in order to process data streams. For

example, in the case of NG, the learning rate is kept constant so that $\varepsilon = \varepsilon_i = \varepsilon_f$, where $\varepsilon_i$ and $\varepsilon_f$ are the initial and final learning rates, respectively.

## Initialization

The initialization for OKRB, SOMRB, and NGRB is described in the Algorithms 1, 2, and 3, respectively. For all comparison methods except SOM, the elements of the reference vectors are initialized uniformly at random in the range $[0, 1)$. For SOM, the reference vector is initialized as $\mathbf{w}_n = (\lfloor n/L \rfloor / L, (\mathrm{mod}L)/L, 0, \ldots, 0)$, where $L = \lfloor \sqrt{N} \rfloor$. The units of SOM on the 2D feature map are placed at $p_n = (p_{n1}, p_{n2}) = (\lfloor n/L \rfloor, \mathrm{mod}L)$. Such an initialization strategy strengthens the network topology of the SOM against map distortions. Traditionally, reference vectors are either randomly selected data points or derived using an efficient initialization algorithm such as k-means++ because random initialization often creates dead units. However, in the context of a data stream, data points are continuously fed into the system. Furthermore, their characteristics will change due to concept drift, requiring methods to adapt to the data regardless of the state of the reference vectors. Consequently, any method designed for a data stream must not only extract relevant features from the data, but also strive to minimize the generation of dead units, regardless of how the initial reference vectors are set.

## Evaluation metrics

In this study, the performance of vector quantization algorithms is evaluated using two metrics: the mean squared error (MSE) and the number of dead units $N_{\mathrm{dead}}$. The MSE quantifies the average squared distance between each data point and its nearest unit, expressed as follows

$$\mathrm{MSE} = \frac{1}{M} \sum_{i=1}^{M} \sum_{n=1}^{N} k_{in} \| \mathbf{x}_i - \mathbf{w}_n \|^2, \tag{12}$$

where $M$ is the number of data points. The data point $\mathbf{x}_i$ is assigned to the unit $s$ if $s = \mathrm{argmin}_n \| \mathbf{x}_i - \mathbf{w}_n \|^2$, and as a result $k_{is} = 1$. Otherwise $k_{in} = 0$. The unit $n$ is considered a dead unit if $M_n = \sum_{i=1}^{M} k_{in} = 0$. So $N_{\mathrm{dead}} = |\{n | M_n = 0\}|. |\cdot|$ is the number of elements in a set.

To evaluate the topological properties of the networks generated by SOMRB and NGRB, the average degree and the average clustering coefficient are used. The average degree $\bar{k}$ is the average number of edges per unit and can be expressed as

$$\bar{k} = \frac{2L}{N}, \tag{13}$$

where $L$ is the total number of edges and $N$ is the number of units. The average clustering coefficient $C$ is defined as $C = \frac{1}{N} \sum_{n=0}^{N} c_n$, where $c_n$ is the clustering coefficient of the unit $n$. The clustering coefficient $c_n$ is calculated as $c_n = \frac{2t_n}{k_n(k_n - 1)}$, where $t_n$ is the number of triangles around the unit $n$ and $k_n$ is the number of edges formed by the unit $n$. If $k_n < 2$, then $c_n = 0$.

The RB updating frequency indicates the number of RB updating occurrences. It is a candidate metric for concept drift detection because RB updating is assumed to occur in response to dynamic changes in a data stream caused by concept drift.

**Table 4** Characteristics of datasets.

| Dataset | N | D | STD | Max | Min |
|---|---|---|---|---|---|
| Blobs | 1,000 | 2 | – | – | – |
| Circles | 1,000 | 2 | – | ~1 | ~−1 |
| Moons | 1,000 | 2 | – | ~2 | ~−1 |
| Aggregation | 788 | 2 | 9.44 | 36.6 | 1.95 |
| Compound | 399 | 2 | 8.69 | 42.9 | 5.75 |
| t4.8k | 8,000 | 2 | 147 | 635 | 14.6 |
| t7.10k | 10,000 | 2 | 170 | 696 | 0.797 |
| Iris | 150 | 4 | 1.97 | 7.9 | 0.1 |
| Wine | 178 | 13 | 216 | 1,680 | 0.13 |
| digits | 1,797 | 64 | 6.02 | 16 | 0 |

## Software

The implementation of OKRB, SOMRB, NGRB, online k-means, SOM, NG, and GNG was done in Python using several Python libraries including NumPy, NetworkX, and scikit-learn. NumPy facilitated linear algebra computations, while NetworkX aided in network manipulation and network coefficient computations. Scikit-learn was used to generate synthetic data. The source code used in this study is publicly available and can be found at https://github.com/KazuhisaFujita/RemoveBirthUpdating.

## Datasets

In this study, six synthetic datasets and three real-world datasets are used to evaluate the proposed methods. The synthetic datasets include Blobs, Circles, Moons, Aggregation (*Gionis, Mannila & Tsaparas, 2007*), Compound, t7.8k (*Karypis, Han & Kumar, 1999*), and t8.8k (*Karypis, Han & Kumar, 1999*). Blobs, Circles, and Moons are generated using the make_blobs, make_circles, and datasets.make_moons functions from the scikit-learn library, respectively. Blobs are derived from three isotropic Gaussian distributions with default parameters for mean and standard deviation. Circles are composed of two concentric circles generated with noise and scale parameters set to 0.05 and 0.5, respectively. Moons are composed of two moon-shaped distributions generated with the noise parameter set to 0.05. Aggregation, Compound, t4.8k, and t7.10k serve as representative synthetic datasets commonly used to evaluate clustering methods. In addition to the synthetic datasets, three real-world datasets from the UCI Machine Learning Repository are used, including Iris, Wine, and Digits. Table 4 details the characteristics of each dataset, showing the number of data points ($N$), dimensions ($D$), standard deviation (STD), maximum values (MAX), and minimum values (MIN). Note that the STD, MAX, and MIN are not explicitly defined for Blobs, Circles, and Moons due to the random nature of data point generation.

## RESULTS

The numerical experiments were performed to evaluate the performance and explore the features of OKRB, SOMRB, and NGRB. All experimental values are the average of 10 runs

with random initial values. An input vector is uniformly and randomly selected from a dataset at each iteration during training.

## Comparison of network structures from synthetic datasets

Figure 2 shows the distributions of the reference vectors of OKRB, SOMRB, NGRB, online k-means, SOM, NG, and GNG for synthetic datasets; Blobs, Circles, Aggregation, Compound, t4.8k, and t8.8k. The reference vectors are trained by each algorithm over $5 \times 10^4$ iterations. An input is a data point randomly selected from the dataset at each iteration. In this experiment, the random generator uses the same seed. These visualizations provide insight into the structure of the reference vectors and the generated networks.

The figure shows that OKRB, SOMRB, NGRB, and GNG successfully extract the topologies of all datasets. While the SOM-generated network contains some dead units, most of its reference vectors accurately capture the topologies of the datasets. The SOM network topology is a two-dimensional lattice, which often leads to the creation of dead units between clusters. In contrast, online k-means and NG tend to generate dead units when the initial position of the units is significantly away from the dataset. While this dead unit problem could be mitigated by refining the initial value, such a solution is not feasible for data streams, given their unbounded nature and the potential for changing ranges of values in the dataset. This suggests that methods other than online k-means and NG may be more appropriate for dealing with data streams.

## Evaluating method performances on static datasets

Figure 3 shows the evolution of the Mean Squared Error (MSE) over iterations for the static datasets: Blobs, Circles, Aggregation, Compound, t4.8k, t7.10k, Iris, Wine, and Digits. At each iteration, MSE is calculated between all data points in the dataset and the current reference vectors. All four methods (OKRB, SOMRB, NGRB, and GNG) can maintain low MSEs from the $10^4$ iterations. In particular, the MSEs of OKRB and NGRB decay rapidly. The performance of OKRB and NGRB is equal to that of GNG and better than other methods. The MSE of SOMRB is slightly larger than those of OKRB, NGRB, and GNG, but smaller than that of SOM. Despite the lack of parameter decay and possibility to generate dead units, SOM shows a respectable performance. On the other hand, both NG and Online k-means show bad performance, mainly due to the lack of optimal initial reference vector values and a step decay mechanism. However, for the Circle dataset, NG and Online k-means perform well because the initial values are within the data distribution (the range of values for Circle is from $-1$ to $1$).

Figure 4 shows the evolution of dead units over iterations for the static datasets. At each iteration, the number of dead units is calculated between all data points in the dataset and the current reference vectors. OKRB and NGRB keep the number of dead units close to zero from the $10^4$ iterations. SOMRB's rate of decrease in dead units is slower than OKRB and NGRB. Interestingly, despite the 2-dimensional lattice bias and the lack of parameter decay, SOM generates a relatively limited number of dead units. However, compared to methods that use RB updating, the number of dead units produced by SOM is higher. For the Blobs, Circles, Aggregation, t4.8k, t8.8k, and Digits datasets, the number of dead units

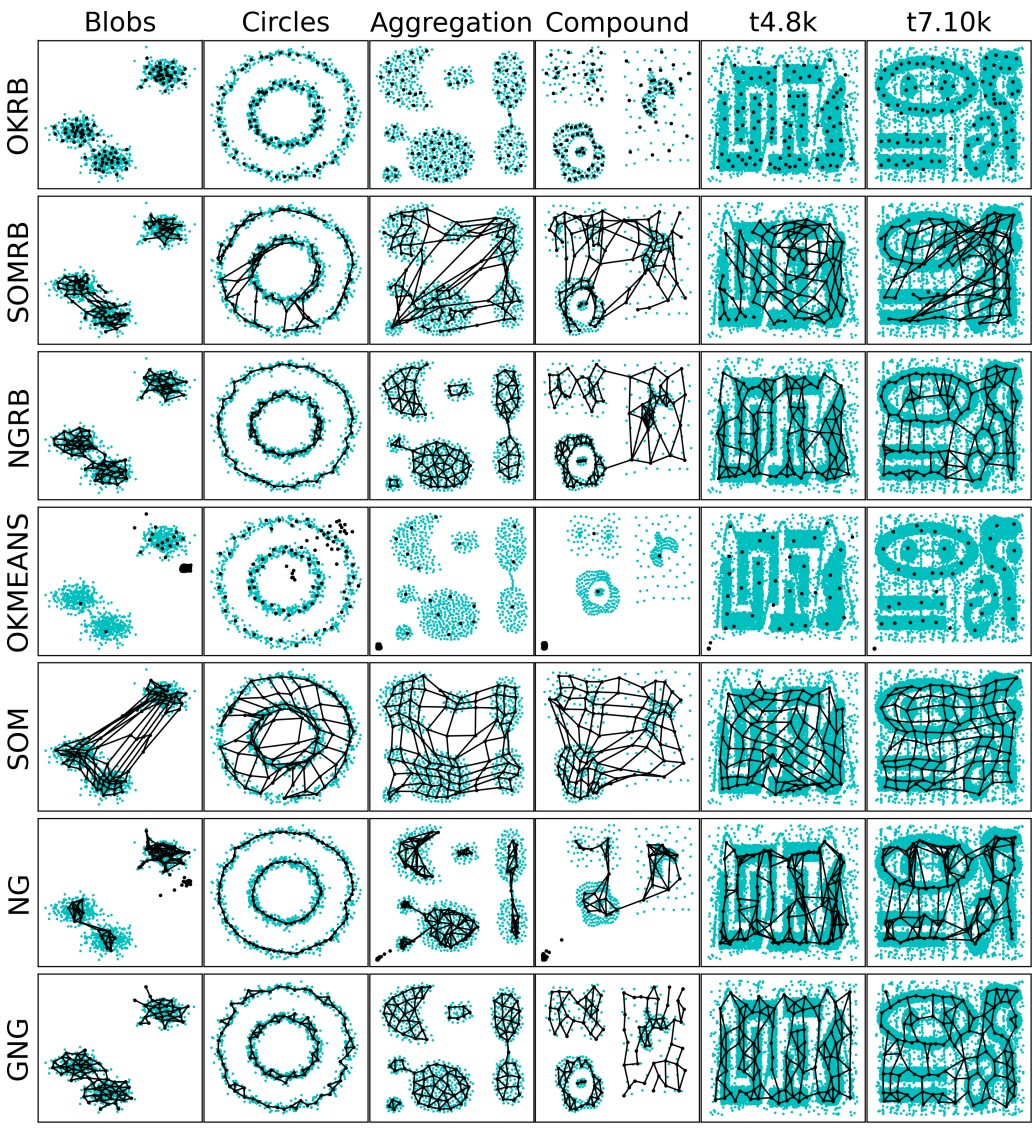

**Figure 2** **2D plots of reference vectors and distributions of datasets.** The first, second, third, fourth, fifth, sixth, and seventh rows show reference vectors generated by OKRB, SOMRB, NGRB, online k-means (OKMEANS), SOM, NG, and GNG, respectively. The data points are represented by cyan dots, while the black dots denote units. The black lines symbolize the edges of the networks.

generated by GNG is approximately zero across all iterations. Conversely, NG and Online k-means have a higher number of dead units and a slower rate of decay of the number of dead units.

These results suggest that the proposed methods, namely OKRB, SOMRB, and NGRB, show sufficient performance with static data. However, due to the constraints of the network topology, SOMRB and SOM show relatively inferior performance compared to OKRB and NGRB. Online k-means and NG, in the absence of parameter decay and efficient initialization algorithms, provide inferior results even with static data. It is suggested that

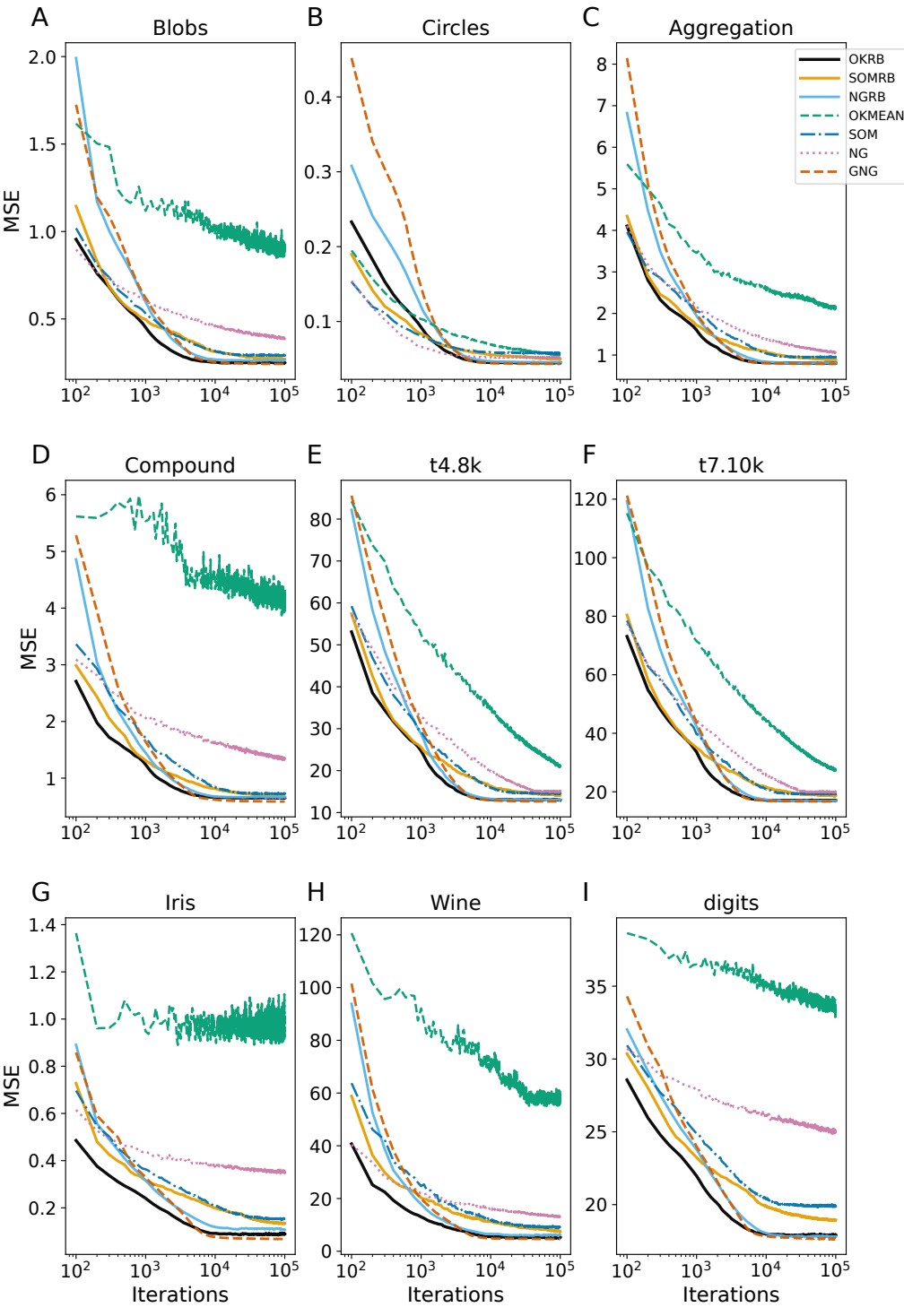

**Figure 3   Iteration evolution of mean squared error (MSE) for various methods for static datasets.** The horizontal axis represents the training iteration, while the vertical axis represents the MSE. Each panel represents a different dataset: (A) Blobs, (B) Circles, (C) Aggregation, (D) Compound, (E) t4.8k, (F) t7.10k, (G) Iris, (H) Wine, and (I) Digits.

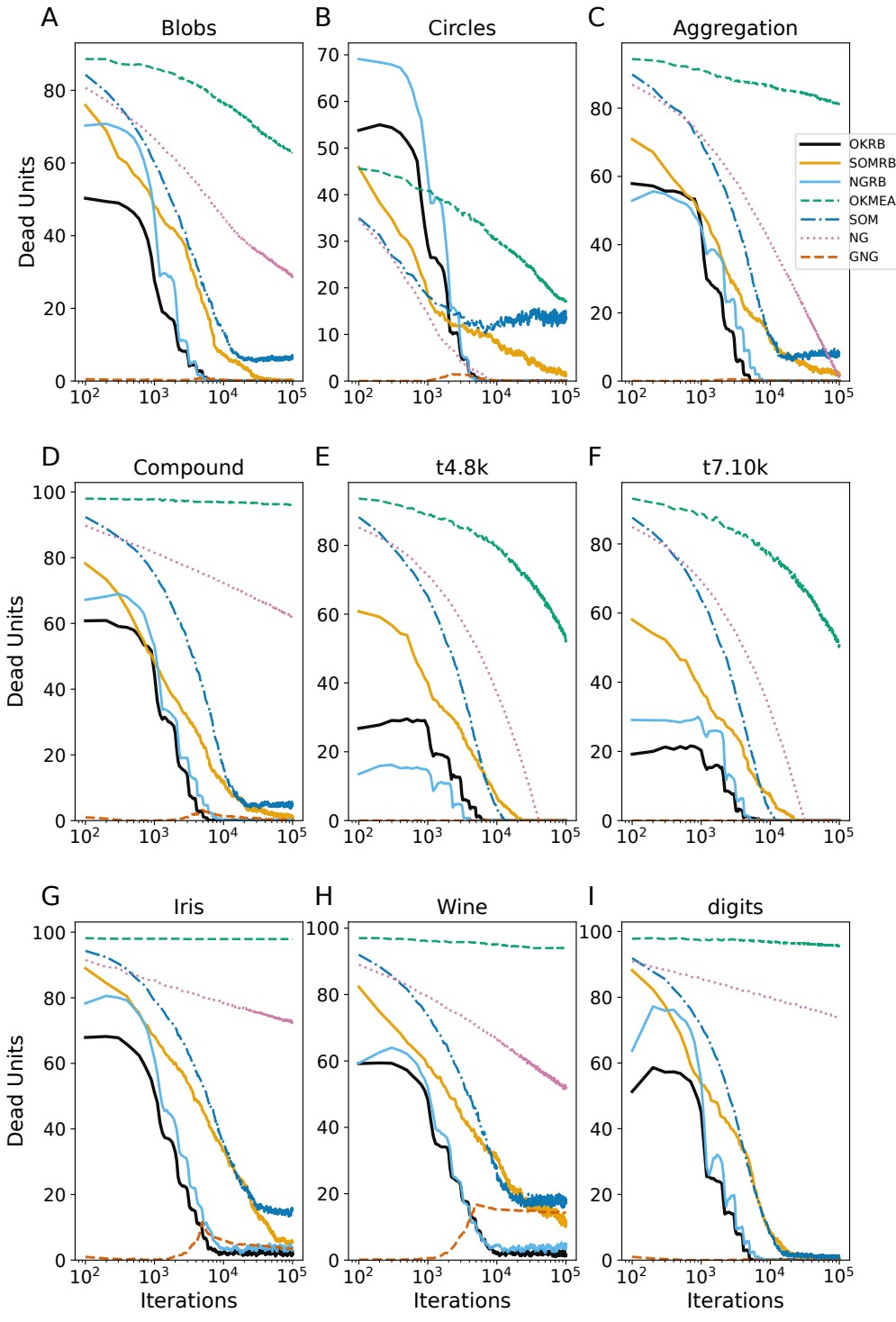

**Figure 4** **Iteration evolution of the number of dead units for various methods for static datasets.** The horizontal axis represents the training iteration, while the vertical axis represents the number of dead units. Each panel represents a different dataset: (A) Blobs, (B) Circles, (C) Aggregation, (D) Compound, (E) t4.8k, (F) t7.10k, (G) Iris, (H) Wine, and (I) Digits.

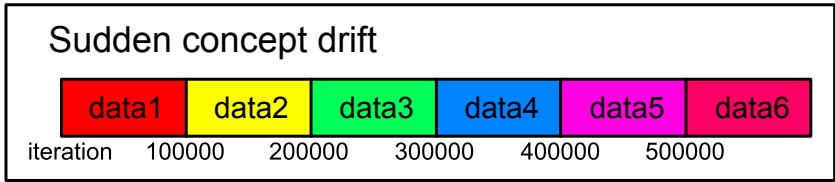

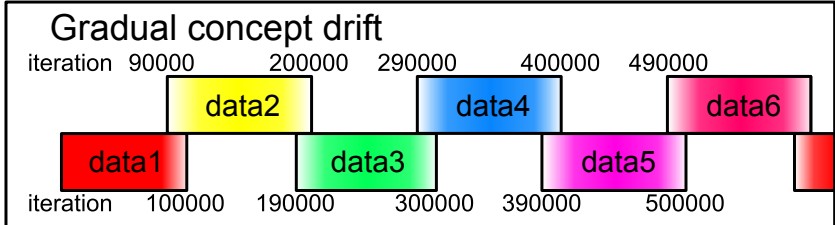

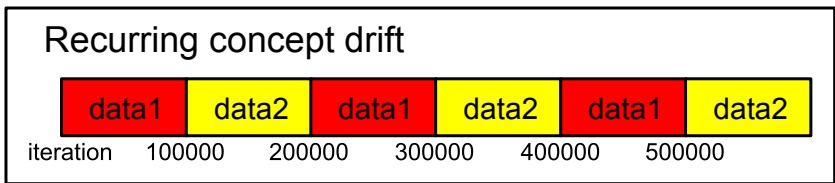

**Figure 5** **Illustration of the data streams used in this experiment.** Data1 through data6 correspond to the aggregation, blobs, circles, compound, t4.8k, and t7.10k datasets, respectively. For both sudden and recurring concept drifts, the drifts occur every 100,000 iterations. In the case of gradual concept drift, a gradual transition from one dataset to another is implemented over the same iteration interval. During this drift, the probability of data generation gradually shifts.

online k-means and NG without decay parameters are also likely unsuitable for stream data.

## Performance for data stream

In this subsection, the proposed methods with RB updating are evaluated for data streams. We consider three types of concept drifts: sudden, gradual, and recurring concept drifts, as shown in Fig. 5. The data stream consists of several different datasets. A concept drift event, which occurs every 100,000 iteration, causes a change from one dataset to another.

For sudden concept drift, the dataset responsible for generating the input undergoes an abrupt change. For gradual concept drift, the dataset generating the input gradually shifts from the old dataset to the new dataset. To elaborate, at each iteration, a dataset generating an input is probabilistically selected, and an input is uniformly and randomly chosen from the selected dataset. The selection probabilities of the old and the new datasets are represented by $p_{old} = (T_{driftstart} + T_{dur} - t)/T_{dur}$ and $p_{new} = 1 - p_{old}$, where $t$ is the number of iterations, $T_{dur}$ is the duration of the drift, and $T_{driftstart}$ is the start iteration of the drift. In this experiment we set $T_{dur} = 10000$ and the initial $T_{driftstart}$ to 90,000. For the recurring concept drift, two datasets alternately generate the input, switching every 100,000 iteration.

The experiments in this subsection compute the Mean Squared Error (MSE), the number of dead units, the average degree, and the average clustering coefficient. At each iteration $t$, the MSE and the number of dead units are calculated using the data points collected

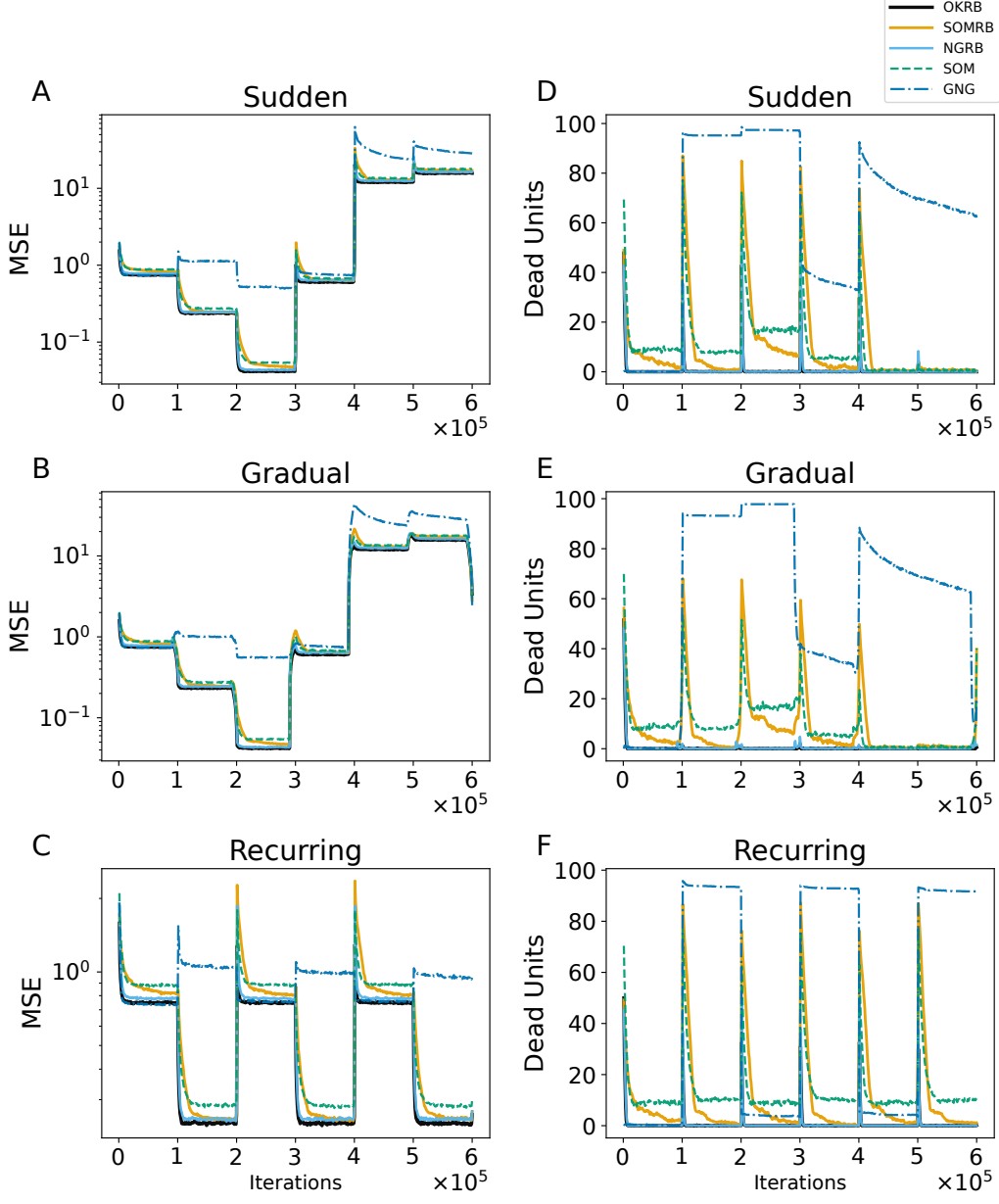

**Figure 6** (A, B, C) The evolution of the mean squared errors (MSE) under sudden concept drift, gradual concept drift, and recurring concept drift scenarios, respectively. (D, E, F) The evolution of the number of dead units under the same scenarios: sudden, gradual, and recurring concept drifts, respectively. The horizontal axis shows the training iteration. The vertical axis shows MSE in A–C and the number of dead units in D–F.

from iteration steps $t - 1000$ to $t$ and the current reference vectors. Additionally, the RB updating frequency at iteration $t$ is defined as the number of RB updating occurrences from iteration steps $t - 1000$ to $t$.

Figures 6A, 6B, and 6C show the MSE evolution of OKRB, SOMRB, NGRB, SOM, and GNG for all types of concept drifts. OKRB, SOMRB, NGRB, and SOM quickly converge to

low MSE values. However, the MSE of SOM is consistently higher than that of the proposed method with RB updating. Although GNG converges quickly to a low MSE value after the arrival of the first stream, it converges to a high MSE value after any concept drift. Figure 6C shows a rapid convergence of GNG's MSE between steps 200,000 and 300,000. This rapid convergence is due to the fact that the third dataset active during this interval is identical to the first. Therefore, GNG maintains its reference vectors that adapt to data1 after the first concept drift. Examples of the reference vector distributions of OKRB, SOMRB, and NGRB during gradual concept drift can be found in 'Evolution of the reference vectors' of the Appendix.

Figures 6D, 6E, and 6F illustrate the progression of dead units for OKRB, SOMRB, NGRB, SOM, and GNG. In all scenarios, the dead units of OKRB and NGRB quickly converge to zero after the concept drift. SOMRB's dead units decrease more slowly than OKRB and NGRB. SOM's dead units decrease more slowly than the proposed methods, and its amount also converges higher than the proposed methods. For sudden concept drift and gradual concept drift, OKRB, SOMRB, NGRB, and SOM have few dead units between data5 (t4.8k) and data6 (t7.10k) because Data5 and Data6 are widely distributed with noise. GNG has more dead units than other methods after the first concept drift.

These observations suggest that OKRB, SOMRB, and NGRB deal with concept drift efficiently. Significantly, OKRB, SOMRB, and NGRB are not affected by changes in the value range of the data due to concept drift. This range independence is attributed not only to RB updating but also to a property of online k-means, SOM, and NG, namely the independence of the hyperparameters on the value range of the data. The learning rates of online k-means, SOM, and NG, as well as the parameters of the neighborhood functions of NG and SOM, do not need to be adjusted based on the value range of the data, despite changes in the characteristics of a dataset. For example, we typically do not change the learning rate whether the maximum value of the data is 100 or 1. In addition, the hyperparameters of the neighborhood functions of SOM and NG depend on the location of the units on the feature map and the neighborhood ranking, respectively.

Interestingly, SOM does not show poor performance with concept drift. In general, SOM can only learn static datasets because once a map learns and stabilizes, it loses its ability to reshape itself as new structures manifest in the input data (*Smith & Alahakoon, 2009*). The decay parameters provide this stability by giving the SOM the flexibility to adapt to a static dataset in the early stages of training while maintaining stable reference vectors. On the other hand, the static parameters derive the SOM's flexibility for concept drift because the static parameters provide a continuous learning capability. In addition, when concept drift occurs, and the SOM's reference vectors diverge significantly from the input vectors generated by a data stream with new characteristics, multiple reference vectors are simultaneously adapted to the input vectors through the neighborhood function. As a result, SOM without decay can quickly adapt to the new characteristics and maintain flexible learning even as data characteristics change.

Figures 7A, 7B, and 7C show the evolution of the average degree of SOMRB and NGRB. The average degree at each iteration is calculated from the network obtained at that time. For sudden and gradual concept drifts, the average degree of SOMRB shows a rapid change

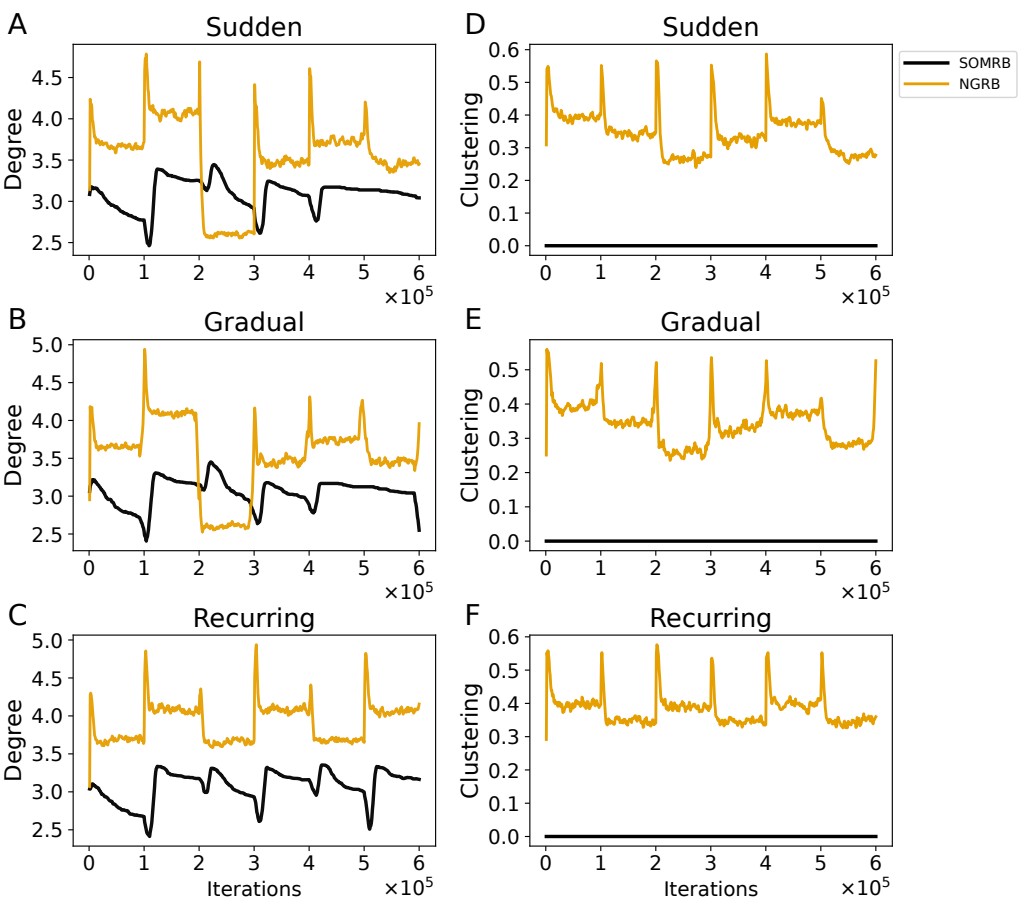

**Figure 7** (A, B, C) The evolution of the average degree under sudden concept drift, gradual concept drift, and recurring concept drift scenarios, respectively. (D, E, F) The evolution of the average clustering coefficient under the same scenarios: sudden, gradual, and recurring concept drifts, respectively. The vertical axis shows the average degree in A–C and the average clustering coefficient in D–F.

at each drift event, except for the transition from data5 to data6. It shows a continuous decay that does not stabilize within each period. The average degree of NGRB peaks at each drift event. However, it does not occur during the transition from data2 to data3 in the gradual concept drift. The average degree of NGRB shows a rapid decay and stabilizes within each drift period. Furthermore, its convergence value is different for each dataset.

Figures 7D, 7E, and 7F show the evolution of the average clustering coefficient of SOMRB and NGRB. The average clustering coefficient at each iteration is calculated from the network obtained at that time. Since the units of SOMRB are placed on vertices of the 2D grid, they cannot form triangular clusters, resulting in a clustering coefficient of zero. On the other hand, NGRB's clustering coefficient shows a peak at each drift, except for the transition from data5 to data6 for the gradual concept drift. Similar to its degree, NGRB's clustering coefficient decays rapidly and stabilizes within each drift period. Furthermore, these convergence values are different for each dataset.

While the value ranges for data5 (t4.8k) and data6 (t7.10k) are similar, and the MSEs for these data are also similar, there are differences in the convergence values of the average degree and the average clustering coefficient. These results suggest that these two metrics can effectively identify shifts in data stream characteristics (concept drift occurrences) when using SOMRB and NGRB.

Figure 8 shows the evolution of RB updating frequency in OKRB, SOMRB, and NGRB. A strong peak in update frequency is observed coinciding with the onset of each concept drift. However, the peaks for the gradual concept drift are lower than for the other drifts. In particular, the peak observed during the transition from data5 to data6 shows a significant reduction. Figures 8D, 8E and 8F show that the peak shapes for OKRB and NGRB are almost identical. Conversely, the peaks for SOMRB are lower in height and broader in width than those for OKRB and NGRB. In addition, the peaks for SOMRB are delayed from the onset of drift. These findings suggest the potential of RB update frequency as an effective indicator for detecting the occurrence of concept drift.

The proposed methods may also be useful for drift detection. As shown in Fig. 6, both the Mean Squared Error (MSE) and the number of dead units increase significantly with concept drift. Similarly, Fig. 7 shows significant changes in the average degree and clustering coefficient when concept drift occurs. Furthermore, these two measures show different values for each data characteristic. Figure 8 shows that the frequency of RB updates shows spike-like fluctuations in response to concept drift. Therefore, these measures could be effectively used to detect concept drift. Moreover, if we use these measures and the proposed methods multiply and simultaneously, we will be able to detect drift even more accurately.

Figures 9A, 9B, and 9C show the evolution of the MSEs for OKRB, SOMRB, and NGRB using the error-based metric. For details on RB updating using the error-based metric, see 'RB updating using error-based metric' in the Appendix. The MSEs of the methods using the error-based metric show rapid convergence. However, for gradual concept drift, the MSEs of OKRB and NGRB using the error-based metric are larger than those of OKRB and NGRB using the win probability based metric during the data4, data5, and data6 phases.

Figures 9D, 9E, and 9F show the evolution of the number of dead units of OKRB, SOMRB, and NGRB using the error-based metric. In all tested scenarios, the number of dead units fluctuates significantly. Furthermore, these methods show a larger number of dead units when using the error-based metric compared to the win probability based metric. Strikingly, NGRB shows an exceptionally high number of dead units for gradual concept drift.

These results suggest that the proposed methods using the win probability based metric demonstrate proficiency in dealing with concept drift. In addition, they significantly mitigate the occurrence of dead units.

## CONCLUSIONS AND DISCUSSIONS

In this study, we proposed three improved vector quantization methods using RB updating for data streams (OKRB, SOMRB, and NGRB). These proposed methods demonstrate fast

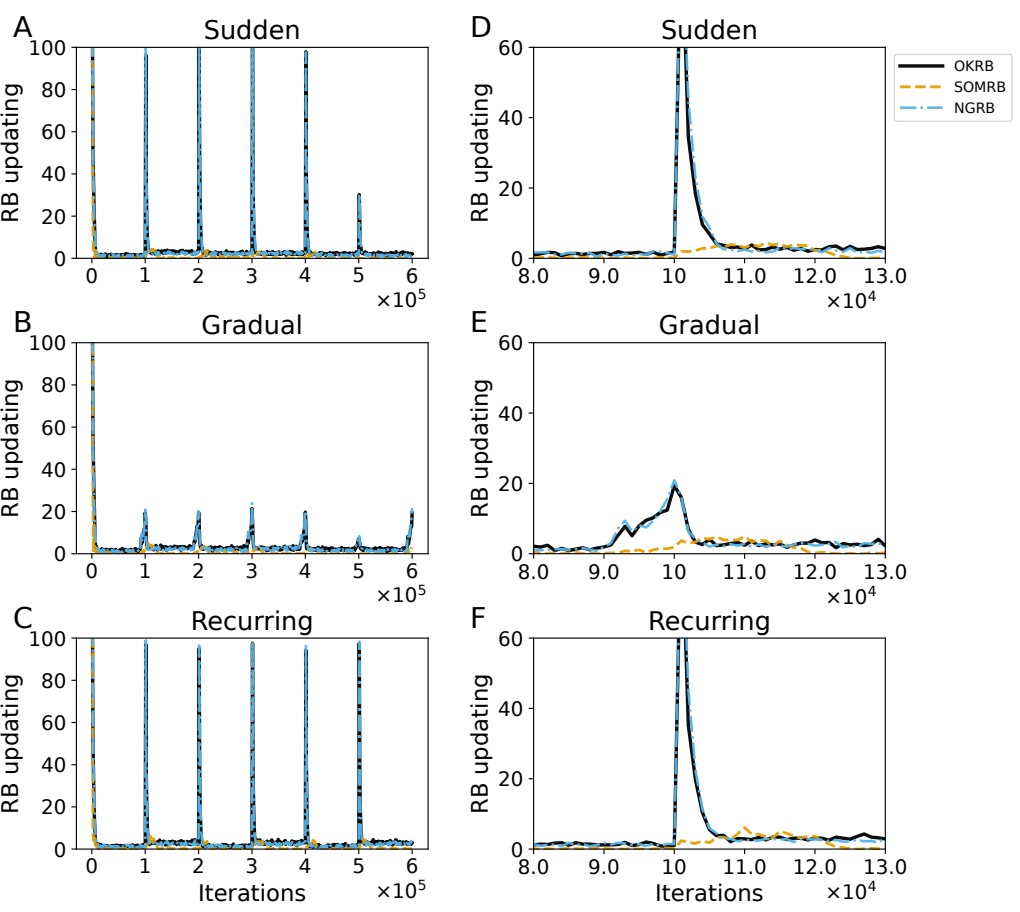

**Figure 8** (A, B, C) The evolution of the frequency of RB updating in OKRB, SOMRB, and NGRB under sudden, gradual, and recurring concept drifts, respectively. (D, E, F) Close-up views of RB updating occurrences within the iteration range of 80,000 to 130,000. These close-up views reveal the intricate behavior of the RB updating occurrences during this specific interval. The horizontal axis and the vertical axis represent training iteration and the frequency of RB updating, respectively.

adaptability to concept drift and provide efficient quantization of a dataset. In addition, both SOMRB and NGRB can generate a graph that reflects the topology of the dataset. However, the performance of SOMRB is slightly inferior to the other proposed methods. Therefore, OKRB is recommended when only vector quantization is required for a data stream. If the task requires not only quantization but also graph generation from a data stream, NGRB is a more suitable option. SOMRB and SOM can be used to quantize data streams and project them into two-dimensional space. SOM is particularly useful when a network structured as a two-dimensional grid is required.

Why is win probability effective in dealing with concept drift? The answer lies in its dependency of a metric on the range of values in the data. The error-based metric depends on the Euclidean distance between two points. This means that the range of values in the data strongly influences the metric. In addition, it can become more sensitive when

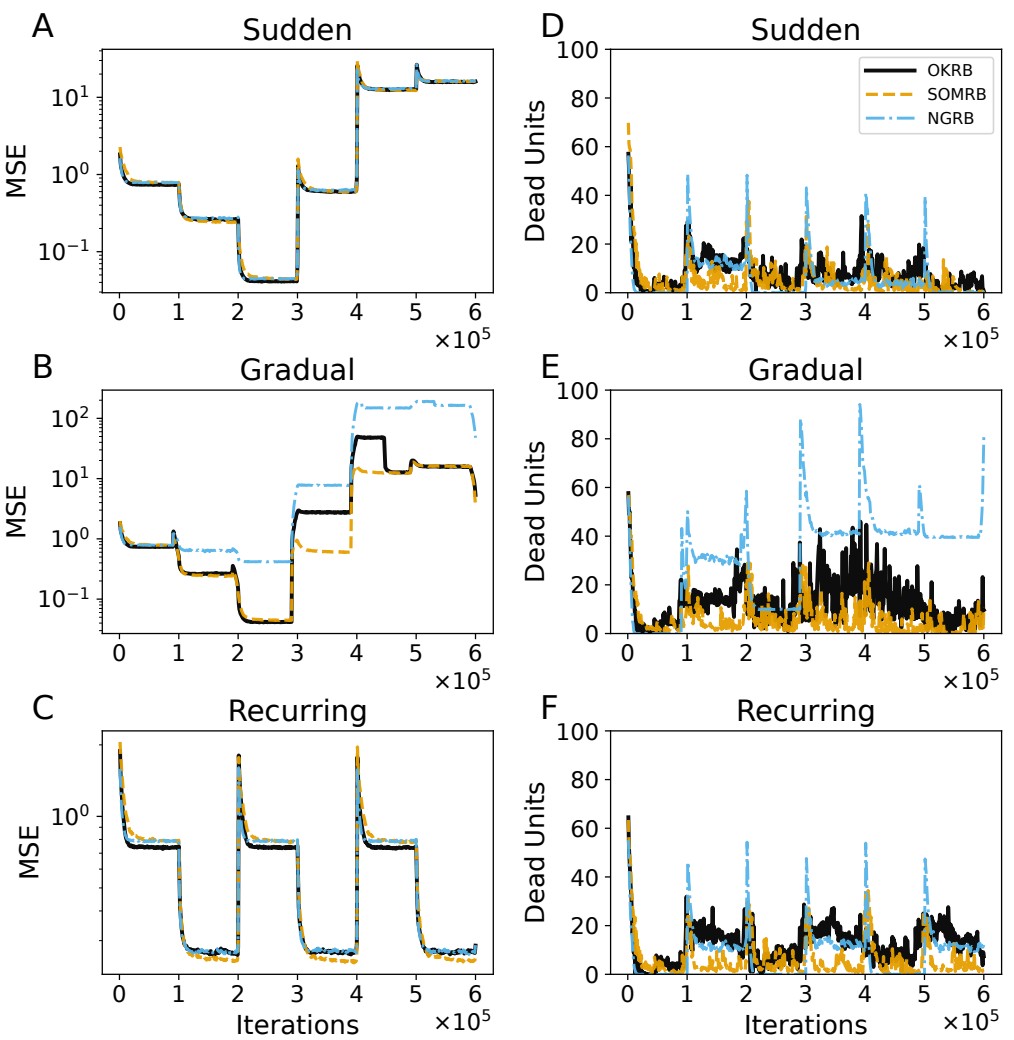

**Figure 9** (A, B, C) The evolution of the Mean Squared Errors (MSEs) of OKRB, SOMRB, and NGRB using an error-based metric under the conditions of sudden, gradual, and recurring concept drifts, respectively. Correspondingly, (D, E, F) The number of dead units under the same drift scenarios: sudden, gradual, and recurring concept drifts. The horizontal axis shows the training iteration. The vertical axis shows the MSE in A–C and the number of dead units in D–F.

dealing with high-dimensional data. This problem is known as the curse of dimensionality. Conversely, a win probability based metric is independent of the range of data values. It works consistently, no matter how spread out the data values are. Thus, a win probability based metric is suitable for concept drift.

Our future work will focus on investigating the effectiveness of the proposed method in a two-step approach that includes an approximate clustering method (*Vesanto & Alhoniemi, 2000*; *Fujita, 2021*). In the first step, a dataset is transformed into sub-clusters. In the second step, these sub-clusters are treated as individual objects and combined into larger clusters. This approach is known to be effective in handling a large dataset (*Vesanto & Alhoniemi, 2000*; *Fujita, 2021*) and a data stream *Mousavi et al.*

*(2020)*. In the first step of this approach, the reduction of dead units is critical because dead units potentially become outliers. In data mining, outliers can negatively affect processing accuracy, making outlier detection a key aspect of the field (*Zubaroğlu & Atalay, 2021*). In cases where concept drift occurs, data points produced by the data stream with old properties and the centroids derived from them could become outliers. The proposed methods quickly reduce the dead units caused by concept drift (*i.e.,* they quickly adapt to the new properties of a data stream), thereby efficiently preprocessing the data stream with new properties. We hypothesize that using our proposed methods in the first sub-clustering step will improve the performance of two-step approximate clustering for the dynamic nature of data streams. We will evaluate this in future work.

# APPENDIX

## Online k-means

Online k-means is an online variant of k-means. This study uses an algorithm derived from the one described in *Abernathy & Celebi (2022)*. For data streams, the online k-means used in this study does not consider the time decay of the parameter. The pseudocode for implementing online k-means can be found in Algorithm 4.

---

**Algorithm 4** Online k-means

---

**Require:** $X = x_1, \ldots, x_t, \ldots, N$

1: **Initialize:**

2:      $w_n = (\xi_1, \ldots, \xi_d, \ldots, \xi_D), \text{n}=1, \ldots, N$, where $\xi_d = [0,1)$ is uniformed random

3: **loop**

4:      {Update reference vectors:}

5:      Receive input $x_t$ at iteration $t$

6:      $s = \arg \min_n \|x_t - w_n\|$

7:      $w_s \leftarrow w_s + \varepsilon (x_t - w_s)$

8: **end loop**

---

## Parameter optimization

In this study, grid search is used for hyperparameter optimization. As a fundamental method for hyperparameter tuning, grid search is both easy to implement and widely accepted (*Bergstra & Bengio, 2012*; *Feurer & Hutter, 2019*).

For grid search, the parameter sets for OKRB, SOMRB, NGRB, online k-means, SOM, NG, and GNG are detailed in Table 5. For each combination of parameters, these methods quantize three different datasets ten times: Blobs, Circles, and Moons.

The goal of the grid search is to find the parameter set that minimizes the normalized mean square error (NMSE), which is computed as follows

$$\text{NMSE} = \sum_{m \in M} \frac{\text{MSE}_m}{\text{MSE}_{\max, m}}, \tag{14}$$

**Table 5  Sets of parameters.**

| OKRB | |
|---|---|
| $\varepsilon$ | 0.05, 0.1, 0.2, 0.3 |
| $Th_{RB}$ | 0.01, 0.05, 0.1, 0.5 |
| $\beta$ | 0.005, 0.001, 0.0005, 0.0001 |

| SOMRB | |
|---|---|
| $\varepsilon$ | 0.05, 0.1, 0.2, 0.3 |
| $\sigma$ | 0.5, 0.75, 1, 2 |
| $Th_{RB}$ | 0.01, 0.05, 0.1, 0.5 |
| $\beta$ | 0.005, 0.001, 0.0005, 0.0001 |

| NGRB | |
|---|---|
| $\lambda$ | 0.5, 1, 2, 4 |
| $\varepsilon$ | 0.05, 0.1, 0.2, 0.3 |
| $A_{max}$ | 25, 50, 75, 100 |
| $Th_{RB}$ | 0.01, 0.05, 0.1, 0.5 |
| $\beta$ | 0.005, 0.001, 0.0005, 0.0001 |

| ok-means | |
|---|---|
| $\varepsilon$ | 0.05, 0.1, 0.2, 0.3, 0.4, 0.5 |

| SOM | |
|---|---|
| $\varepsilon$ | 0.05, 0.1, 0.2, 0.3, 0.4 |
| $\sigma$ | 0.5, 0.75, 1, 2, 3 |

| NG | |
|---|---|
| $\lambda$ | 0.5, 1, 2, 4 |
| $\varepsilon$ | 0.05, 0.1, 0.2, 0.3 |
| $A_{max}$ | 25, 50, 75, 100 |

| GNG | |
|---|---|
| $\lambda$ | 50, 100, 200 |
| $\varepsilon_w$ | 0.05, 0.1, 0.2 |
| $\varepsilon_{ns}$ | 0.0005, 0.005, 0.05 |
| $\alpha$ | 0.25, 0.5, 1.0 |
| $\beta$ | 0.99, 0.999, 0.9999 |
| $a_{maxs}$ | 25, 50, 100 |

where $M$ is the set of datasets: Blobs, Circles, Moons. The MSE for a dataset $m$ is calculated as follows

$$\mathrm{MSE}_m = \frac{1}{N_m} \sum_{i=1}^{N_m} \min_n \|\boldsymbol{x}_i - \boldsymbol{w}_n\|, \tag{15}$$

where $N_m$ is the number of data points in the dataset $m$, and $\boldsymbol{x}_i$ is a data point in the dataset $m$. $\mathrm{MSE}_{\max,m}$ is the maximum MSE derived from all possible parameter combinations for a given dataset $m$.

## Evolution of the reference vectors

Figures 10, 11 and 12 show the evolution of the reference vectors in OKRB, SOMRB, and NGRB, respectively, along with the distribution of the generated data points during the gradual concept drift (transition from Aggregation to Blobs). This drift phase occurs within

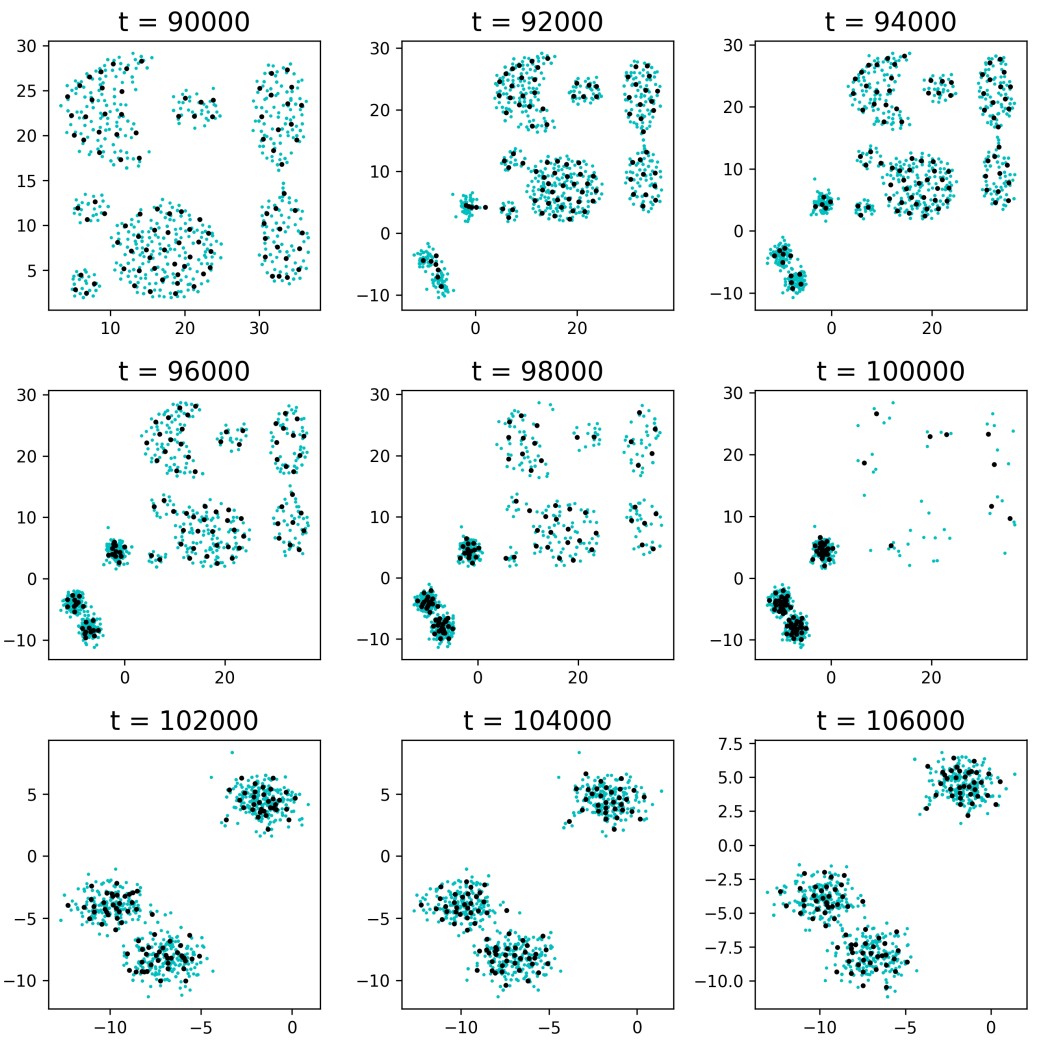

**Figure 10 Scatter plot of data points and corresponding OKRB reference vectors.** Data points are represented by cyan circles, while reference vectors are denoted by black circles.

the time step range of $t = 90000$ to $t = 100000$. The plotted data points are generated within the time span of $t - 1000$ to $t$.

During this gradual concept drift, all three methods successfully capture the topological changes and integrate the old and new features of the data. In particular, after the drift, the reference vectors of OKRB and NGRB, representing the old dataset, quickly disappear. In contrast, for SOMRB, the reference vectors representing the old dataset decrease more slowly.

### RB updating using error-based metric

Each unit has an associated error term, $E_n$, initialized to 0. At each iteration, the error term of the winning unit, denoted $E_{s_1}$, is updated according to the following formula:

$$E_{s_1} \leftarrow E_{s_1} + \|\boldsymbol{x}_t - \boldsymbol{w}_{s_1}\|^2, \tag{16}$$

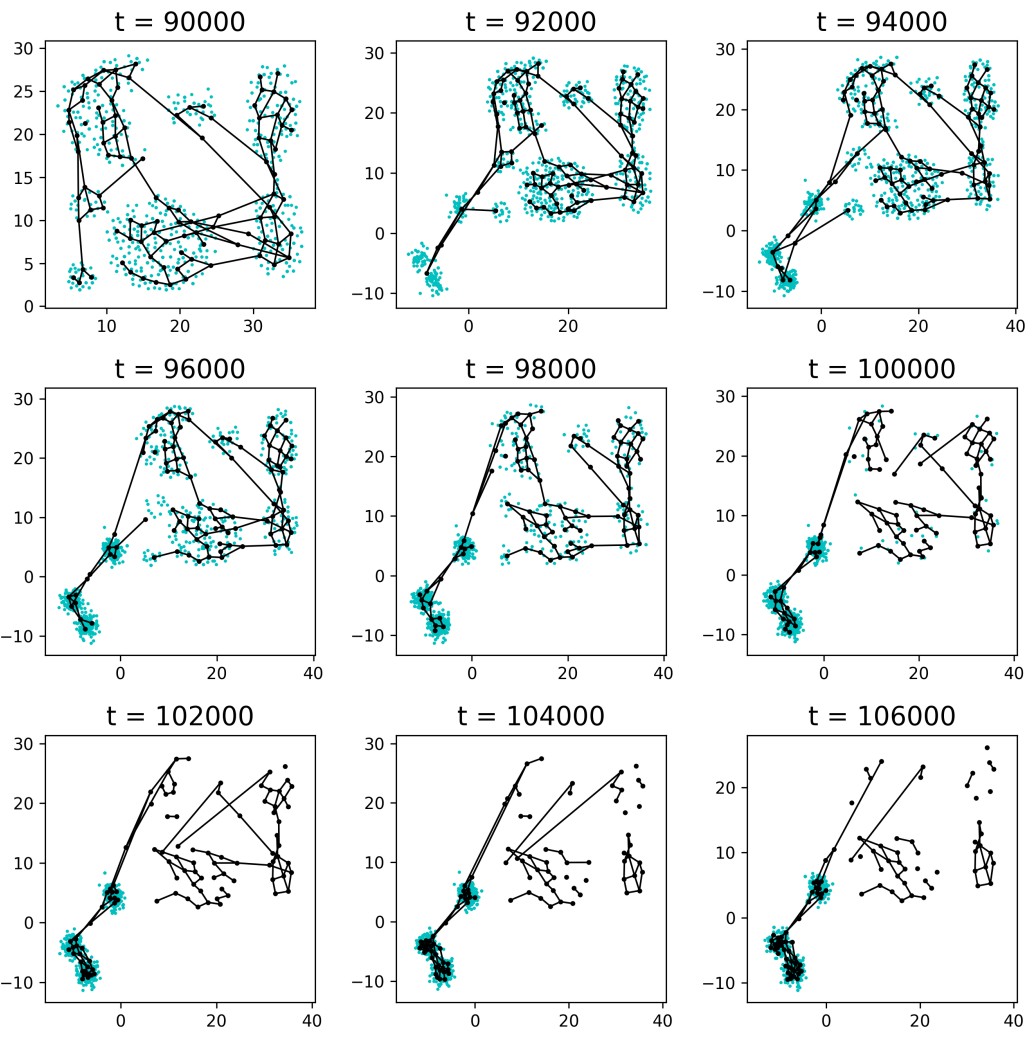

**Figure 11 Scatter plot of data points and corresponding SOMRB reference vectors.** Data points are represented by cyan circles, while reference vectors are denoted by black circles. Edges are denoted by black solid lines.

where $\boldsymbol{x}_t$ is the input vector and $\boldsymbol{w}_{s_1}$ is the reference vector of the winning unit. The error term $E_n$ for all units undergoes a decay process described by the equation:

$$E_n \leftarrow E_n - \beta E_n, \tag{17}$$

where $\beta$ is the decay rate. In addition to the error term, each unit is also associated with a utility term, $U_n$, which is initialized to 0. At each iteration, the utility of the winning unit, $U_{s_1}$, is updated according to:

$$U_{s_1} \leftarrow U_{s_1} + \|\boldsymbol{x}_t - \boldsymbol{w}_{s_2}\|^2 - \|\boldsymbol{x}_t - \boldsymbol{w}_{s_1}\|^2, \tag{18}$$

where $s_2$ is the second nearest unit of the input vector $\boldsymbol{x}_t$. Like the error term, the utility $U_n$ decays at each step according to the following equation:

$$U_n \leftarrow U_n - \beta U_n. \tag{19}$$

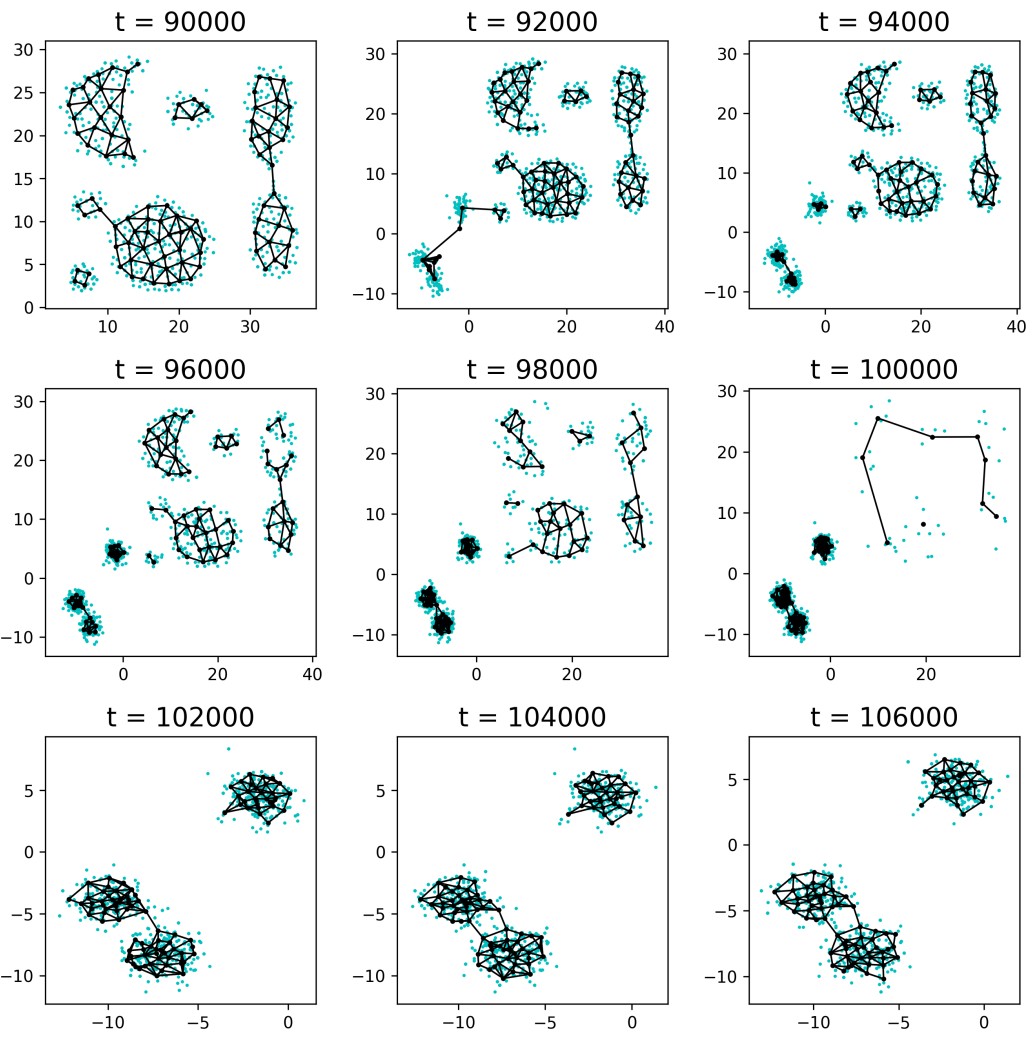

**Figure 12  Scatter plot of data points and corresponding NGRB reference vectors.** Data points are represented by cyan circles, while reference vectors are denoted by black circles. Edges are denoted by black solid lines.

In RB updating with an error-based metric, a unit $n$ with minimum utility $U_n$ is removed and a new unit is created near the unit $q$ with maximum error $E_q$ if the following conditions are satisfied:

$$\frac{U_n}{E_q} < \text{TH}_{\text{RB}}. \tag{20}$$

### Funding

The authors received no funding for this work.

## Competing Interests

The authors declare there are no competing interests.

## Author Contributions

- Kazuhisa Fujita conceived and designed the experiments, performed the experiments, analyzed the data, performed the computation work, prepared figures and/or tables, authored or reviewed drafts of the article, and approved the final draft.

## Data Availability

The codes and the below datasets are available at Github and Zenodo:

- https://github.com/KazuhisaFujita/RemoveBirthUpdating

- Kazuhisa Fujita. (2023). KazuhisaFujita/RemoveBirthUpdating: New release (release). Zenodo. https://doi.org/10.5281/zenodo.10076879

The datasets, Blobs, Circles, and Moons were generated using the 'make_blobs', 'make_circles', and 'datasets.make_moons' functions, respectively, from the Scikit-learn library (https://scikit-learn.org/stable/modules/classes.html#module-sklearn.datasets).

The Aggregation and Compound datasets are available at: Available at https://cs.joensuu.fi/sipu/datasets/. The T7.8k and t8.8k datasets are available at Github: https://github.com/milaan9/Clustering-Datasets/blob/master.

The real-world datasets such as Iris (Fisher, https://archive.ics.uci.edu/dataset/53/iris), Wine (https://archive.ics.uci.edu/dataset/109/wine), and Digits (https://archive.ics.uci.edu/dataset/80/optical+recognition+of+handwritten+digits) are available at the UCI Machine Learning Repository.

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
