# Peer review of "An efficient and straightforward online vector quantization method for a data stream through remove-birth updating"

_PeerJ Computer Science, doi:10.7717/peerj-cs.1789_

## Round 0.1 · original submission · Minor Revisions

Based on the reviewers’ comments, you may resubmit the revised manuscript for further consideration. Please consider the reviewers’ comments carefully and submit a list of responses to the comments along with the revised manuscript.

Reviewer 1 ·

Basic reporting

This paper studies vector quantization for data streams and proposes remove-birth (RB) based quantization methods to address the concept drift. These methods include online k-means RB (OKRB), self-organizing maps RB (SOMRB), and neural gases RB (NGRB). Finally, evaluations of the proposed methods are conducted on both several synthetic and real-world datasets.

Experimental design

The experimental design is well-organized and the results are clearly listed. One question regarding the evaluation:
-- The proposed methods are compared to the original SOM, NG, and GNG. Variants of SOM and GNG are mentioned in Section 3 (line 141). Can the referred methods serve as a baseline?

Validity of the findings

Requested revisions:

-- Section 3, paragraph starting at line 141: explain the shortcoming of the mentioned methods.

Reviewer 2 ·

Basic reporting

This paper proposes a simple online vector quantization method for concept drift. The proposed method identifies and replaces units with low win probability through remove-birth updating, thus achieving a rapid adaptation to concept drift.

A. Lines # 10 & 21, this is not the proper definition of big data.
B. The first paragraph of the introduction presented important terminologies etc but without literature support.
C. Line # 78, concept drift only related to data stream.
D. There is some content repetition in the last 2 paragraphs of the introduction section.
E. It will be better for the reader's point of view if related work is written in chronological order.
F. Similarly, the related work section should have some conclusion at the end. Discussion of the previously published work without any conclusion is not making too much sense. Normally, related work acts like a foundation stone for the proposed solution
G. Fig. 1 needs to be redrawn with clear boundaries of A, B and C.
H. Algorithms 1,2 and 3 should be explained line by line for the reader's understanding.
I. Parameters section 4.5 can better be represented in a tabular format.
J. Inconsistency in using word dataset and data set.
K. Fig. 3, 4 and 6 use complete pages but still its quality is not too good.
L. The conclusion section is unnecessarily long. It contains repetition. Reference to the figures is mentioned in the conclusion which I have hardly seen before. A suggestion is to reduce the conclusion section and if required then move the text to the other part of the paper.
M. The conclusion section is missing future directions.
N. Enough references have been cited.

Experimental design

Experimental design details are provided, and results can be reproduced because of the source code provision.

Validity of the findings

Different figures are given but how results are generated to draw these figures are missing.

Additional comments

Overall, a good article.
Manuscript formatting recommended for peerj is not followed. For example,
• Left justify all text to the left margin. Do not 'full width' justify.
• Similarly, figures and tables should be uploaded separately.

---

## Round 0.2 · accepted · Accept

Congratulations, the reviewers are satisfied with the revised version of the manuscript and have recommended the acceptance decision.

Reviewer 1 ·

Basic reporting

no comment

Experimental design

no comment

Validity of the findings

no comment

Additional comments

The author has made revisions to the requested changes.

Reviewer 2 ·

Basic reporting

All recommended changes have been made.

Experimental design

Fine.

Validity of the findings

Fine.

Additional comments

Much improved manuscript after revision.